# SIM: Surface-based fMRI Analysis for Inter-Subject Multimodal Decoding from Movie-Watching Experiments

**Simon Dahan**[1][*] **Gabriel Bénédict**[2][*][†] **Logan Z. J. Williams**[1] **Yourong Guo**[1]
**Daniel Rueckert**[3,4,5] **Robert Leech**[6] **Emma C. Robinson**[1]

[1]Research Department of Biomedical Computing, BMEIS, King's College London
[2]Amazon, Madrid
[3]Institute for AI in Medicine, Technical University of Munich
[4]Department of Computing, Imperial College London
[5]Munich Center for Machine Learning (MCML)
[6]Institute of Psychiatry, Psychology & Neuroscience, King's College London

`simon.dahan@kcl.ac.uk, emma.robinson@kcl.ac.uk`

## Abstract

Current AI frameworks for brain decoding and encoding, typically train and test models within the same datasets. This limits their utility for cognitive training (neurofeedback) for which it would be useful to pool experiences across individuals to better simulate stimuli not sampled during training. A key obstacle to model generalisation is the degree of variability of inter-subject cortical organisation, which makes it difficult to align or compare cortical signals across participants. In this paper we address this through use of surface vision transformers, which build a generalisable model of cortical functional dynamics, through encoding the topography of cortical networks and their interactions as a moving image across a surface. This is then combined with tri-modal self-supervised contrastive (CLIP) alignment of audio, video, and fMRI modalities to enable the retrieval of visual and auditory stimuli from patterns of cortical activity (and vice-versa). We validate our approach on 7T task-fMRI data from 174 healthy participants engaged in the movie-watching experiment from the Human Connectome Project (HCP). Results show that it is possible to detect which movie clips an individual is watching purely from their brain activity, even for individuals and movies *not seen during training*. Further analysis of attention maps reveals that our model captures individual patterns of brain activity that reflect semantic and visual systems. This opens the door to future personalised simulations of brain function. Code & pre-trained models will be made available at `https://github.com/metrics-lab/sim`.

## 1 Introduction

Over recent years, there has been growing interest in the extent to which machine learning frameworks, such as convolutional neural networks (CNNs) and transformers, can model neurological processes: from spatial encoding of the hippocampus (Ellwood, 2024; Whittington et al., 2021; Kim et al., 2024) to replicating semantic (Antonello & Huth, 2024; Caucheteux et al., 2023; Huth et al., 2016; Millet et al., 2022) and visual (Benchetrit et al., 2023; Ozcelik & VanRullen, 2023; Tang et al., 2024; Wen et al., 2018) representations within the cortex. These approaches support the testing of new theories of human cognition (Millet et al., 2022; Antonello & Huth, 2024; Caucheteux et al., 2023; Ellwood, 2024; Kriegeskorte, 2015; Whittington et al., 2021; Kim et al., 2024), and allow for the encoding or decoding of stimuli (Benchetrit et al., 2023; Défossez et al., 2023; Gu et al., 2022; Lindsay, 2021; Kriegeskorte, 2015; Ozcelik & VanRullen, 2023; Scotti et al., 2024; Thomas

---

[*]Equal contribution
[†]Work done outside of Amazon

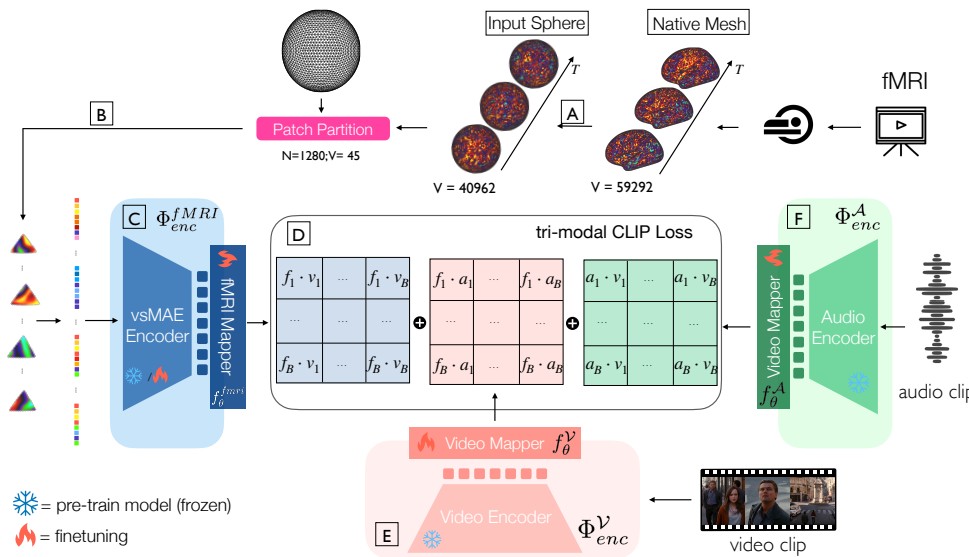

Figure 1: (**SIM**) seeks to align visual and audio stimuli - extracted from $T = 3$ second movie clips - with brain activations acquired over the same time period: [A] 7T fMRI data, collected during movie watching, is first projected to subjects' native cortical surfaces (resolution $V$=59292 vertices); then inflated to a sphere, and downsampled onto a regularly tessellated icosphere ($I_6$ with $V$=40962 vertices). [B] This is then encoded using a surface vision transformer (SiT), which tokenizes data by patching each $I_6$ sphere with a lower-resolution icospheric grid to generate a sequence of triangular patches ($V$=45 vertices per patch; $N$=1280 number of patches). [C] The SiT encoder ($\Phi_{enc}^{fMRI}$) is pre-trained as part of a video surface masked autoencoder (vsMAE) for fMRI-frame reconstruction; [D] and fMRI embeddings ($f_i$) are then aligned with CLIP contrastive training to video ($v_i$) [E] and audio ($a_i$) [F] embeddings learnt from a videoMAE ($\Phi_{enc}^{\mathcal{V}}$) and wav2vec ($\Phi_{enc}^{\mathcal{A}}$) model; and projected to vector spaces of common lenght by multimodal mappers ($f_\theta^{fMRI}, f_\theta^{\mathcal{V}}, f_\theta^{\mathcal{A}}$). At test time this makes it possible to decode video/audio stimuli from fMRI (or vice versa) through comparing the similarity of their CLIP embeddings ($f_i, a_i, v_i$). This comparison generates a probability distribution over the candidate samples, allowing for evaluation of retrieval performance using top-K accuracy metrics.

et al., 2022; Thual et al., 2023; Wen et al., 2018; Yamins & DiCarlo, 2016) from non-invasive human brain recordings such as functional magnetic resonance imaging (fMRI), magneto-encephalography (MEG) or electro-encephalography (EEG). However, such comparative models are subject-specific, typically require large amounts of imaging data per subject, and do not generalise to unseen individuals. This limits the extent to which they can be used to probe individual sources of variation, to ultimately build precision models of human cognition and behaviour.

One contributing factor to the lack of generalisation of current models, is arguably that most models treat signals from adjacent locations in the brain as spatially independent - encoding patterns of brain activity using linear, or non-linear regression modules (Huth et al., 2016; Millet et al., 2022; Caucheteux et al., 2021; Scotti et al., 2024; Ozcelik & VanRullen, 2023). This ignores the well-documented spatial auto-correlation of signals across the cortex that is known to be behaviourally meaningful (Bijsterbosch et al., 2018; Kong et al., 2019; Shinn et al., 2023; Leech et al., 2023; Margulies et al., 2016).

At a high level, cortical activity may be decomposed into signals from a relatively small number of functionally specialised areas (Glasser et al., 2016a; Smith et al., 2013). These regions communicate through neuronal connections, resulting in coordinated patterns (or networks) of activity. Previous studies have shown that human brain function may be modelled from temporal sequences of these networks (Vidaurre et al., 2017; Pervaiz et al., 2022; Smith et al., 2013). This suggests that if we can build an encoding model that generalises these spatial patterns across individuals, then we can predict what any individual's cortical functional dynamics should be in response to a given stimulus; or predict a novel stimulus from the spatial topography of their brain activity.

With this in mind, recent efforts to generalise vision transformers (ViTs) (Dosovitskiy et al., 2020) to the cortical surface present significant opportunity as they allow for the modelling long-range spatio-temporal interactions through the encoding of space-time self-attention (Dahan et al., 2022; 2024). Relative to a 3D network of equivalent resolution, surface models are much more compact and require less memory; this creates scope for building more sophisticated network architectures. Moreover, surface modelling more faithfully represents the true geometry of the convoluted cortical surface, and for this reason has long been favoured for analysis of cortical signals (Glasser et al., 2016b; Margulies et al., 2016; Gordon et al., 2017b;a).

**Contributions:** In this paper, we introduce **SIM** (Fig 1), a novel framework that combines self-supervised contrastive learning with surface vision transformers (SiT) (Dahan et al., 2022; 2024), to support *multimodal* decoding of audio+video (movie) stimuli from 7T cortical fMRI. Importantly, this generalises to predicting *new* movie clips from *new* individuals (not used during training); using only a few seconds of fMRI; and is trained and tested using the HCP 7T movie-watching dataset (Van Essen et al., 2013; Finn & Bandettini, 2021), which collects many hours less data per individual, than most recent decoding frameworks (Benchetrit et al., 2023; Défossez et al., 2023; Scotti et al., 2024; 2023; Ozcelik & VanRullen, 2023). Achieving this involved engineering contributions for tri-modal CLIP alignment of audio, video stimuli with fMRI. Benefits extend to reconstruction, where our embeddings may be used to reconstruct movie frames, from the brain activity of unseen subjects, that reliably decode the semantic content of scenes. Results show that joint encoding of audio/video stimuli and fMRI inserts complementary information that allows better decoding between any pair of these modalities. Furthermore, visualisation of self-attention suggests the model attends to functional brain networks in audio-visual information. This opens the door to the future precision simulation (Digital Twins) of human brain function in response to unseen stimuli and tasks.

## 2 RELATED WORKS

Classic approaches to fMRI decoding, classify stimuli from the temporal dynamics of voxels/vertices extracted from visual regions of the human brain. Most often linear or non-linear regression is used; with recent models leveraging the power of generative or foundational AI to learn rich encodings of stimuli, which are subsequently matched to patterns of brain activity through use of CLIP contrastive learning (Benchetrit et al., 2023; Défossez et al., 2023; Scotti et al., 2024; 2023; Ozcelik & VanRullen, 2023). Often methods are complemented with "hyper-alignment" techniques that seek to map all data to a space in which brain activations overlap across individuals (and so may compared) (Haxby et al., 2020; Thual et al., 2023); however, in practice, this is ill-posed[1] which has meant that most decoding frameworks (Benchetrit et al., 2023; Défossez et al., 2023; Scotti et al., 2024; 2023; Ozcelik & VanRullen, 2023) are trained and tested within the same brain, using densely sampled datasets (Allen et al., 2021; Hebart et al., 2023) that collect tens of hours of recordings for each subject. Similarly, approaches to inter-subject generalisation (Scotti et al., 2024; Thual et al., 2023) introduce costly alignment steps that require $\geq 1$ hour of recording for each test subject.

Learning-based frameworks support transformation equivariant modelling of images and, as such, offer improved potential for generalisation. However, the cortex is a highly curved manifold – with activations best modelled as patterns across a surface (Glasser et al., 2016b;a; Coalson et al., 2018). This presents challenges due to the difficulty in translating convolutional operations to non-Euclidean domains that lack a global coordinate system (Bronstein et al., 2021); resulting in frameworks that perform sub-optimally on cortical phenotyping and fMRI decoding tasks (Fawaz et al., 2021; Gu et al., 2022).

Recent work on SiTs (Dahan et al., 2022) has indicated that vision transformers robustly outperform surface convolutional neural networks (CNNs) across a range of cortical phenotyping tasks, while offering inherent interpretability through visualisation of self-attention. Moreover, careful adaption of the video masked autoencoder pre-training (Tong et al., 2022; Feichtenhofer et al., 2022) to surface domains was shown to encode sufficient rich representations of cortical dynamics to decode cognitive traits (Dahan et al., 2024).

---

[1]due to noise and limited temporal resolution

## 3 METHODS

In what follows, we consider cortical fMRI signals as functions in space and time $S(v, t)$ defined on a spherical mesh ($v \in V_6$) corresponding to a 6th-order icosahedron: $I_6 = (V_6, F_6)$, with $|V_6| = 40962$ vertices and $|F_6| = 81920$ faces, see Figure 1.A. The objective of our framework is to use SiTs to encode the spatio-temporal dynamics of the signal as it evolves. This is implemented through video surface-masked autoencoder (vsMAE) self-supervision. Decoding is then implemented through aligning representations learnt from the vsMAE (Fig 1.C) with audio and visual representations (learnt from wav2vec (Baevski et al., 2020) and videoMAE (Tong et al., 2022)) using CLIP contrastive learning (Fig 1.D). At inference time, this supports the retrieval of any one modality from each of the others.

### 3.1 BASE ARCHITECTURES

**SiT:** Cortical surface analysis Fischl (2012), is now a very common processing stage for most neuroimaging data collections, involving the fitting of tessellated meshes to the inner and outer boundaries of the cortex, followed by inflation and projection to a sphere (Figure 1.A). Cortical imaging features, such as fMRI are projected onto the surface through ribbon-constrained weighted averaging (Glasser et al., 2013). The SiT (Dahan et al., 2022) leverages this simplified spherical domain to patch cortical imaging data using regularly tessellated icosahedral meshes. First imaging features are resampled to a high-resolution $I_6 = (V_6, F_6)$ mesh (with $|V_6| = 40962$ vertices and $|F_6| = 81920$ faces; these features are then patched with the faces of a low-resolution icosphere (typically $I_3$, with $|F_3| = 1280$) (Fig 1.B) - partitioning the cortical features into a sequence of $N = |F_3|$ non-overlapping triangular patches: $P = \{p^1, p^2, ..p^{|N|}\}$ (with $p^i \subset V_6, |p^i| = 45$). Features within each patch are then concatenated across channels ($C$) flattened and projected, with a trainable linear layer, into a set of $D$-dimensional input tokens to produce an initial sequence: $X^0 = [X_1^0, ..., X_N^0] \in \mathbb{R}^{N \times D}$. Sine-cosine positional embeddings, $E_{pos} = \{E_i\}_{i=1}^N$ are then added to each of the tokens to encode patch location within the sequence: $\mathcal{X}^{(0)} = [X_1^0 + E_1, ..., X_N^0 + E_N]$. This initial sequence $\mathcal{X}^{(0)}$ is then processed by $L$ consecutive transformer encoder blocks of $H$ *Multi-Head Self-Attention* (MHSA) and *Feed Forward Network* (FFN) layers, with residual layers in-between, resulting in an output sequence of fMRI token embeddings ($\mathcal{X}_{fMRI} \in \mathbb{R}^{N \times D}$). As with classic vision transformers (Dosovitskiy et al., 2020) the objective is to model long-range co-occurrences of spatial structure in the surface imaging data (as self-attention between tokens), which should confer sufficient image understanding to perform any given supervised learning task.

**vsMAE:** Transformers are powerful learning models but their lack of inductive biases present challenges when training on limited data. In Dahan et al. (2024), a self-supervision pre-training task was proposed that extends the concept of video-masked autoencoders (Tong et al., 2022; Feichtenhofer et al., 2022) to the surface. This frames self-supervision as an auto-encoding task, where *unmasked* tokens, from $T$ input fMRI frames, are first randomly selected from the set of all available patches according to a masking ratio $\rho$ . Each token then represents a 'tube' of cortical data patched in space and time. These are passed to a SiT encoder ($\Phi_{enc}^{fMRI}$), which compresses each spatio-temporal token through its linear layer, and then passes the resulting sequence of tokens through each transformer encoder block. Next, random embeddings are added into the encoder's latent embeddings sequence - in place of the masked tokens - restoring the sequence to its original length. Positional embeddings are added to encode spatial information and the resulting sequence is fed into the SiT decoder ($\Phi_{dec}^{fMRI}$). The last layer performs a linear projection to restore the input patch resolution ($T \times |p^i|$). Following He et al. (2021), the vsMAE is optimised by calculating the mean square error (MSE) between the masked input feature patches and their reconstructed versions only. The pre-trained vsMAE encoder ($\Phi_{enc}^{fMRI}$) then forms the basis of the proposed decoding framework (Fig 1).

### 3.2 DECODING NETWORK

Taking learnt representations of the fMRI from the pre-trained vsMAE encoder ($\mathcal{X}_{fMRI}$), the next objective is to align these to video ($\mathcal{V}$) and audio ($\mathcal{A}$) representations, respectively noted $\mathcal{X}_\mathcal{V} \in \mathbb{R}^{N_\mathcal{V} \times D_\mathcal{V}}$ and $\mathcal{X}_\mathcal{A} \in \mathbb{R}^{N_\mathcal{A} \times D_\mathcal{A}}$, learnt from pre-trained videoMAE (Tong et al., 2022) and wav2vec2.0 models (Baevski et al., 2020) ($\Phi_{enc}^\mathcal{V}$ and $\Phi_{enc}^\mathcal{A}$ in Fig 1.E & F), using CLIP contrastive

learning (Radford et al., 2021). Details about audio-visual stimuli processing and embeddings extraction are provided in Appendix B.3.

**Multimodal mappers:** Since each unimodal model outputs a sequence with different token length, it is necessary to first compress all representations to vectors of the same length such that they can be directly compared (Fig 1.D). This is performed with multimodal mappers $f_\theta^{mod}$, similar to (Najdenkoska et al., 2023), that project each unimodal sequence of tokens through two *linear* layers, with *GeLU* activation, dropout and residual connections, before averaging the sequence to output a vector of equivalent resolution ($D_{CLIP}$) for each modality: such that $y_{fMRI} = f_\theta^{fMRI}(\mathcal{X}_{fMRI})$, $y_\mathcal{A} = f_\theta^\mathcal{A}(\mathcal{X}_\mathcal{A})$ and $y_\mathcal{V} = f_\theta^\mathcal{V}(\mathcal{X}_\mathcal{V})$, and $y_{fMRI}, y_\mathcal{V}, y_\mathcal{A} \in \mathbb{R}^{D_{CLIP}}$.

**Alignment:** Tri-modal CLIP alignment is then achieved by sampling, for each batch, exactly one positive triplet and $M-1$ negative triplets. A positive triplet consists of fMRI, audio, and video data from the same $3s$ movie clip, while negative triplets are sampled from different $3s$ movie clips. Next, cosine similarities ($z_{a,b}(i,j) = \langle y_a^i, y_b^j \rangle$) are calculated between pairs of modalities ($a$ and $b$), which are then converted into probabilities through applying a softmax function: $P_{a,b}(i,j) = \frac{\exp(z_{a,b}(i,j)/\tau)}{\sum_{k=1}^M \exp(z_{a,b}(i,k)/\tau)}$ (here, $\tau$ is a temperature hyperparameter that scales the logits). The CLIP loss from modality $a$ to $b$ (noted $L_{a \to b}$) is then calculated using cross-entropy to push together the embeddings of positive samples and push apart the embeddings of negative samples, such that: $L_{a \to b} = -\frac{1}{M} \sum_{i=1}^M \log P_{a,b}(i,j)$. To perform alignment across three modalities simultaneously, we average the losses calculated between all pairs of audio/video/fMRI in the batch:

$$L = (L_{f_{MRI} \to \mathcal{V}} + L_{\mathcal{V} \to f_{MRI}} + L_{f_{MRI} \to \mathcal{A}} + L_{\mathcal{A} \to f_{MRI}} + L_{\mathcal{A} \to \mathcal{V}} + L_{\mathcal{V} \to \mathcal{A}})/6$$

Then, at inference time, and for each given modality $a$, the model can be used to rank the samples that are most likely to be aligned with modality $b$ (highest probability) from a list of available stimuli.

**Reconstruction:** Using the methodology proposed by (Ozcelik & VanRullen, 2023) (and employed in (Benchetrit et al., 2023)), we developed a two-stage framework to reconstruct visual stimuli from cortical fMRI data. In the first stage, a regression model ($reg_1$) is trained to predict visual latent representations (extracted from a pre-trained VDVAE model (Child, 2021)) from fMRI signals. In the second stage, a separate regression model ($reg_2$) is trained to map fMRI CLIP embeddings ($y_{fMRI}$) to video CLIP embeddings ($y_\mathcal{V}$). This model is trained on one fMRI session for one training subject. During inference, test fMRI signals (from an *new* subject and *new* movie scene) are first processed by $reg_1$ to generate latent visual representations and decoded by the VDVAE model to produce low-resolution image reconstructions. These low-resolution images, along with multimodal guidance from $reg_2$, are input into a pre-trained Versatile Diffusion (VD) model (Rombach et al., 2022) to generate the final high-resolution visual reconstructions, see Figures 5, C.10 and C.11.

# 4 EXPERIMENTAL METHODS

## 4.1 DATASET

In this paper, stimuli and accompanying brain recordings were taken from 174 participants, aged $29.4 \pm 3.3$ years (68 male and 106 female) who were scanned as part of the HCP 7T movie-watching experiment (Van Essen et al., 2013; Finn & Bandettini, 2021). Participants underwent 4 recording sessions, each lasting $\sim$ 15 minutes, during which time they were shown a series of movie scenes from independent/Hollywood movies (1-4.3 mins in length). These movie scenes were interleaved with rest periods of 20 seconds, where participants were told to fixate on a cross on a blank screen (Fig 2). For each session, audio was delivered via earbuds, and movie files were cropped and zoomed from their original 16:9 aspect ratio (AR) to a $1024 \times 720$, 14.22:10 AR to fit the projector screen. All Hollywood movie scenes were prepared and published by (Cutting et al., 2012).

During sessions, fMRI was acquired on a 7 Tesla Siemens Magnetom scanner, using a gradient-echo EPI sequence with repetition time (TR) = $1s$, echo time (TE) = $22.2ms$, and spatial resolution = $1.6mm^3$ (Finn & Bandettini, 2021). These volumes were motion- and distortion-corrected, high-pass filtered, and boundary-based aligned (Greve & Fischl, 2009) to T1- and T2-weighted sMRI acquired at $0.7mm^3$ resolution, by the HCP's custom 3T Siemens Skyra (Glasser et al., 2013).

Confound signals were then regressed from the data, where these correspond to 24 measured msotion parameters (at each timepoint) and a series of data-driven noise components derived from FIX ICA (Salimi-Khorshidi et al., 2014)).

All data was processed through the HCP minimal processing surface pipelines (Glasser et al., 2013) (adapted from FreeSurfer (Fischl, 2012)). This maps fMRI timeseries onto tessellated meshes representing each individual's cortical anatomy. Data is then mapped to the sphere and aligned using MSMAll functional alignment, where this has been shown to considerably improve the overlap of functional activations across brains, relative to volumetric or cortical shape-based alignment (Robinson et al., 2014; 2018; Glasser et al., 2016a; 2013; Smith et al., 2013; Coalson et al., 2018). Features are resampled from native resolution (59292 vertices) to $|I_6|$, using adaptive barycentric interpolation (Robinson et al., 2018).

## 4.2 TRAINING

Subjects were partitioned into train/validation/test splits of size 124/25/25, while stratifying sex and age distribution across splits, with fMRI from left and right hemispheres treated as independent samples but placed in the same split. This corresponds to 992 training, 200 validation and 200 testing samples. We then divide the movie data into a series of non-overlapping $3s$ movie clips: corresponding to 16 frames of movie stimuli and 3 frames from the cortical fMRI, where this was sampled with a temporal lag of 6 seconds to account for the haemodynamic response (Huth et al., 2016) (Appendix C.8).

The SiT backbone, forming $\Phi_{enc}^{fMRI}$, follows the standard structure of a DeiT-small (Touvron et al., 2020), which extends (Dosovitskiy et al., 2020) to more efficient ViT architectures. For all training phases (vsMAE pre-training and tri-modal CLIP alignment), the AdamW (Loshchilov & Hutter, 2019) optimisation was used with $LR = 3e^{-4}$ and cosine decay. Distributed training was implemented in all experiments with a batch size of 64 (per GPU) for the vsMAE pre-training task. For the tri-modal CLIP training, batch size was maximised across all GPU instances to 256 by implementing aggregation across instances[2]. All experiments were run on 4 NVIDIA V100 GPUs (32 GB of memory). Video and audio encoders ($\Phi_{enc}^{\mathcal{V}}$, $\Phi_{enc}^{\mathcal{A}}$) were kept frozen for all experiments. Multimodal mappers ($f_\theta^{fMRI}$, $f_\theta^{\mathcal{V}}$, $f_\theta^{\mathcal{A}}$) were trained in all experiments. During the tri-modal CLIP training, we investigated various training regimes for the pre-trained vsMAE encoder: (1) training from scratch, (2) keeping the SiT encoder frozen after vsMAE pre-training or (3) fine-tuning from pre-trained weights. To evaluate the impact of different modalities during the CLIP-alignment training, we also investigated the training of two modalities only (e.g. $f_{MRI}$, $\mathcal{V}$) or the three modalities altogether ($f_{MRI}$, $\mathcal{V}$, $\mathcal{A}$). Ablation experiments for (1) & (2) are shown in Table C.1.

## 4.3 INFERENCE

At test time, models were evaluated based on their retrieval and reconstruction abilities, where inference of modality $a$ from modality $b$ is defined as $a \rightarrow b$ (e.g. $f_{MRI} \rightarrow \mathcal{V}$). In practice, this is calculated from sampling $M$ test samples, including one positive (correct) pair and $M - 1$ negative (mismatched) pairs. These are passed to the CLIP model, which outputs a (softmax) probability that each of these pairs is a match. Results are then ranked and performance is reported from top-$K$ accuracy i.e. whether the true pair ranks within the top $K$ samples, as per recent works (Thual et al., 2023; Ozcelik & VanRullen, 2023; Scotti et al., 2023; 2024; Benchetrit et al., 2023; Défossez et al., 2023). The number of clips tested is adapted for each pair of modalities to account for different noise levels - applying $M = 64$ for video and $M = 32$ for audio testing (audio samples being noisier). Two types of negative sampling procedures were used: *soft-negative* - where negative pairs are sampled only from different movies to the positive sample; and *hard-negative* sampling - where negatives were sampled only from within the same movie (and thus share semantic content). Importantly, we used a sampling buffer around the positive sample, ensuring that no negatives were taken from within $\pm 3s$ of a positive sample. The choice of $3s$ to define movie clips corresponds to the average duration of movies 'shots' (see more details in Appendix A.2). Main retrieval results are provided in Table 1 and Figure C.2 and additional results in Tables C.1, C.3 & C.2 and Figure C.3.

---

[2]Implementation provided in `https://github.com/metrics-lab/sim`

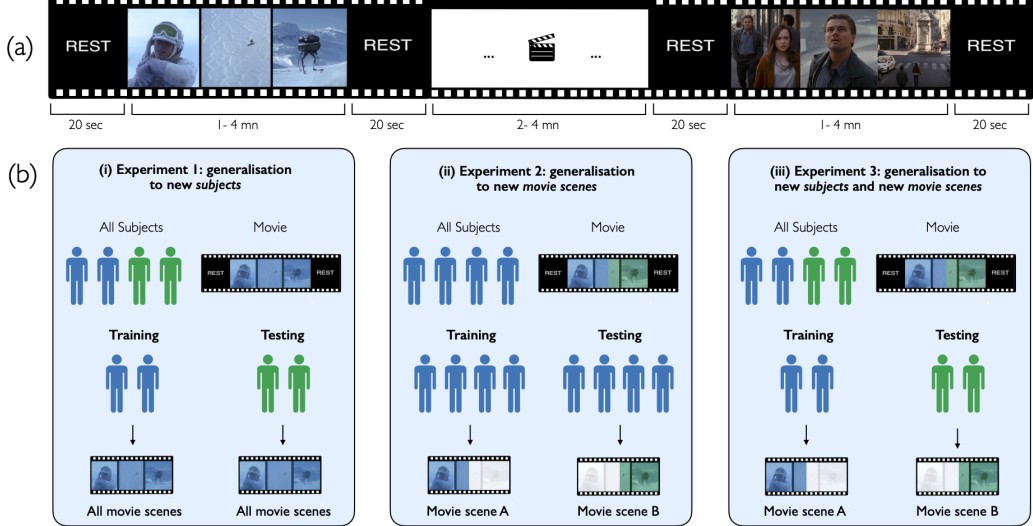

Figure 2: (a) Each movie-watching session (MOVIE1-4) is composed of *movie scenes* extracted from different *movies* (ranging in total from $1.4$ to $4.3mn$) and interleaved with $20s$ rest intervals. Movie scenes are further divided into $3s$ *movie clips* for training and inference processes. (b) Overview of the three experimental setups: (i) *Experiment 1*: Subjects are divided into training and testing groups; training involves all movie clips while testing validates whether we can decode movie clips that were seen during training but using brain activations of *new* subjects from the test set. (ii) *Experiment 2*: utilises all subjects for both training and testing, but only the first half of each movie as training; we then validate on whether we can decode *new* clips from movie scenes that were not seen during training. (iii) *Experiment 3*: Training is limited to a subset of subjects (as in (i)) *and* only the first half of each movie (as in (ii)); the model is then tested on decoding *new* movie clips - from the last half of all movies - from the brain activations of *new* subjects not seen during training. Figure C.1 further clarifies the multilevel sampling terminology.

## 4.4 EVALUATION

The objective of this paper is to demonstrate that it is possible to train encoding and decoding frameworks that generalise to (i) *new* subjects (*Experiment 1*); (ii) *new* movie scenes (*Experiment 2*); and (iii) *new movie scenes* **within** *new subjects* (*Experiment 3*). The experimental setup is described in Figure 2.

Beyond evaluation of retrieval performance, we also consider: (1) how well self-attention maps, encoded by the vsMAE, reflect the current understanding of the underlying cognitive processes; (2) if tri-modal alignment allows stimuli reconstructions from fMRI signals. We, therefore, visualise attention maps by projecting them back to the cortical surface (Fig4). This is achieved by extracting the self-attention-weights matrix from all layers, passing these through softmax operations; then slicing to retain only weights associated with the output $[CLS]$ token (following Caron et al. (2021)). These weights are then interpolated back to $I_6$ resolution by assigning all vertices, for each cortical patch, the attention weight of the corresponding token in the sequence. For the video-frame reconstruction, we adapt the code from Ozcelik & VanRullen (2023) to the CLIP/latent encoding resolutions of our architecture; applying 50 DDIM steps with a strength of 0.75.

**Baseline:** To evaluate the contribution of the SiT over and above the impact of contrastive learning, we compare against Ridge regression models, taking inspiration from Ozcelik & VanRullen (2023). Since all data has been MSMall (functionally) aligned across subjects, this represents an extremely robust baseline.

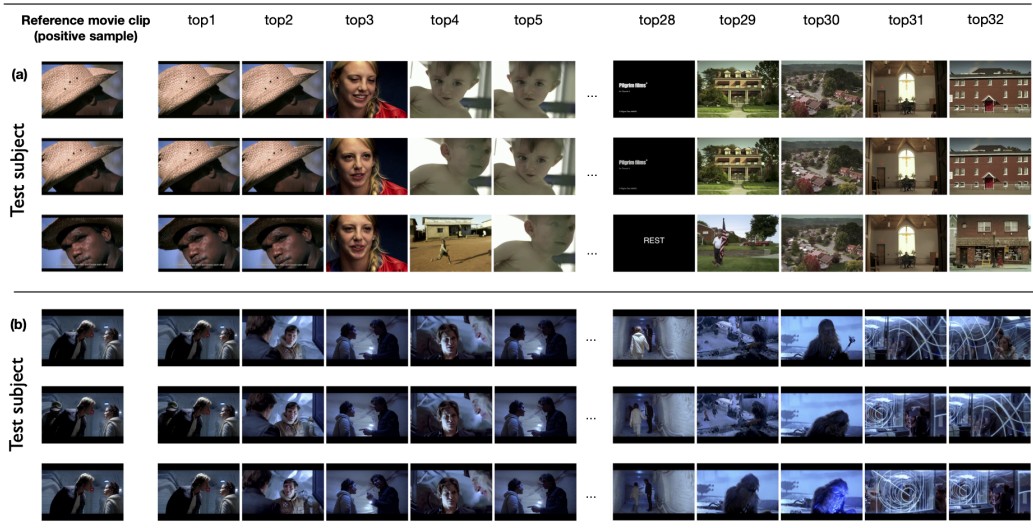

Figure 3: Video retrieval for *Experiment 3* - generalisation to *new movie scenes* and *new subjects* - from: (a) *soft-negative* sampling: here, the reference movie clip is correctly retrieved as top1 and top-ranked movie clips all depict human faces; (b) *hard-negative* sampling: here the top-ranked movie clips all correspond to dialogue scenes.

## 5 RESULTS

**Generalisation to *new* subjects (i)** Results in Table 1 report the top-1 and top-10 performance for *Experiment 1* (described in Fig 2) in two inference directions $f_{MRI} \rightarrow \mathcal{V}$, $f_{MRI} \rightarrow \mathcal{A}$. In both cases, the best retrieval performance is achieved when training on all three modalities, demonstrating the complementary contributions of video and audio for decoding (see also Appendix: Table C.1 for the inverse retrieval results). In all cases, performance increases considerably over random and ridge regression baselines, even for CLIP alignment of only two modalities. Table 1 summarises results for *hard-negative* sampling and decoding only - but comparable improvements are seen for *soft-negative* sampling, and inverse retrieval ($\mathcal{V} \rightarrow f_{MRI}$ and $\mathcal{A} \rightarrow f_{MRI}$), which may be seen as encoding brain activations from audio/video stimuli. For more results, please refer to Appendix C: Table C.1, Figure C.7 and Figure C.8.

**Generalising to new movie scenes and new subjects (ii) & (iii)** Results for *Experiment 2* and *Experiment 3* for *soft-negative* sampling are reported in Figure C.2; and demonstrates that the SIM frameworks clearly generalises to *new* movie scenes and *new* subjects, compared to baselines. Visualisation of retrieved clips (Figure 3), for two test subjects, and for both *hard-negative* and *soft-negative* sampling, shows that our model meaningfully ranks movie clips with similar contents together. The full tables of results are available in Appendix C (Table C.2 and C.3); in all cases the proposed model ($f_{MRI} \rightarrow \mathcal{V}$) strongly outperforms baselines.

Table 1: Generalisation to *new subjects* (*Experiment 1*), tested on all movies. Bold corresponds to the best-performing model for each inference modality. Results (in %) with $\bar{\mu}$ and 95% conf. interval (CI).

| Inference Modality | $\Phi_{enc}^{fMRI}$ | Training modalities (CLIP) | clip retrieval top-1 ± CI | top-10 ± CI |
|---|---|---|---|---|
| $f_{MRI} \rightarrow \mathcal{V}$ | Random | ✗ | 3.7± 0.5 | 19.4±0.9 |
| | Ridge | $f_{MRI}, \mathcal{V}$ | 15.6±1.1 | 64.1±1.6 |
| | SiT (ours) | $f_{MRI}, \mathcal{V}$ | 64.7±2.3 | **95.6±1.4** |
| | SiT (ours) | $f_{MRI}, \mathcal{V}, \mathcal{A}$ | **76.8±2.6** | 94.2±1.5 |
| $f_{MRI} \rightarrow \mathcal{A}$ | Random | ✗ | 3.1±0.4 | 30.1±1.1 |
| | Ridge | $f_{MRI}, \mathcal{A}$ | 3.2±0.4 | 30.5±1.2 |
| | SiT (ours) | $f_{MRI}, \mathcal{A}$ | 19.9±1.3 | 62.5±1.7 |
| | SiT (ours) | $f_{MRI}, \mathcal{V}, \mathcal{A}$ | **56.6±2.5** | **86.9±2.0** |

**Interpretation of self-attention maps** Visualisation of attention maps, corresponding to cortical fMRI, averaged across all test subjects, from a $3s$ movie clip involving dia-

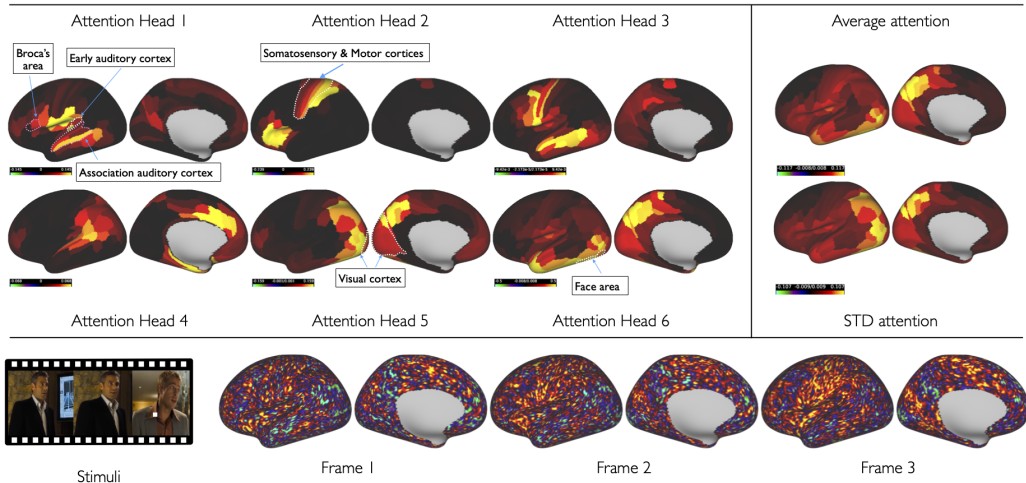

Figure 4: Average attention maps for each attention head, extracted from the SiT encoder ($\Phi_{enc}^{fMRI}$), from a 3s movie clip of a dialogue scene during Ocean's Eleven. Maps are averaged across test subjects (in *Experiment 1*). Brain regions of importance for the movie-watching stimuli are highlighted and annotated based the HCP multimodal parcellation (Glasser et al., 2016b). Scene files will be made available on BALSA. Comparison with functional networks in Appendix C.3.

logue, are shown in Figure 4. Results are shown averaged within the individual HCP multimodal areal parcellation (Glasser et al., 2016b). This is done to support comparisons between cortical areas and their reported functions (see Tables 1-3 supplementary anatomical results (Glasser et al., 2016b)). Results are shown for each attention head separately, we also display the mean and variance calculated across all heads, for all test subjects.

Systematic comparison of the activation patterns from each attention head, against well-validated topographic maps of functional networks (Yeo et al., 2011; Margulies et al., 2016) reveals their specialisation into sensorimotor, visual, and auditory cortices. A correlation analysis against Margulies' gradient-based maps (Margulies et al., 2016) shows that Gradient 2 is the highest correlated with all attention heads, pointing to dominance of attention within visual and sensorimotor/auditory cortices. Closer examination shows that attention heads 1 to 4 specialise in sensorimotor/auditory processing, while heads 5 and 6 specialise in visual processing. Similarly, a comparison with the functional networks identified by Yeo et al. (Yeo et al., 2009) shows that attention heads 1 to 4 highly correlate with Network 2 (sensorimotor and auditory), whereas heads 5 and 6 correlate with Network 1 (visual). Margulies's gradient maps (Margulies et al., 2016) and Yeo's functional networks (Yeo et al., 2009) are provided in Tables C.4 & C.5 for qualitative comparison.

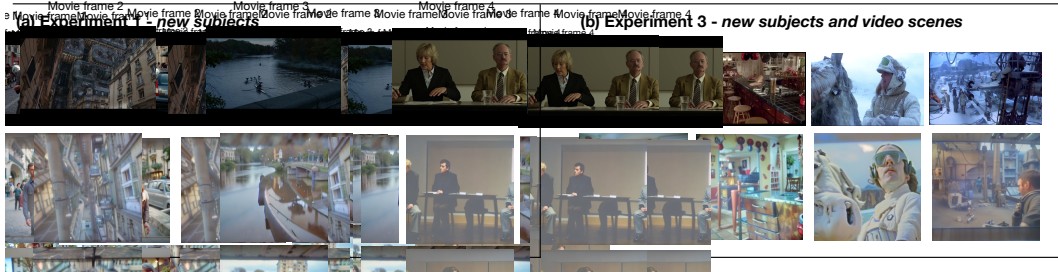

Figure 5: Video-frame reconstruction results from fMRI embeddings after tri-modal CLIP alignment. Following Ozcelik & VanRullen (2023), the reconstruction pipeline was trained on one training subject and tested on a *new subject* - (a) on the same movie scenes as used for training and (b) on new movie scenes. Reconstructions are realistic, preserving most of the semantic information from the original frame and, importantly, generalise to *new movie scenes* and *new subjects*.

## 6 DISCUSSION

Previous studies have shown that modern AI frameworks, such as CNNs and transformers, model natural stimuli in ways that parallel human cognitive processing (Millet et al., 2022; Antonello & Huth, 2024; Caucheteux et al., 2023; Kriegeskorte, 2015) and that this can be harnessed to decode auditory and visual stimuli from human brain recordings (Benchetrit et al., 2023; Défossez et al., 2023; Scotti et al., 2024; 2023; Ozcelik & VanRullen, 2023; Gu et al., 2022; Lindsay, 2021; Thomas et al., 2022; Thual et al., 2023; Wen et al., 2018; Yamins & DiCarlo, 2016). While powerful, these approaches lack any model of individual cortical areal topography from which to simulate unseen stimuli. In this paper we take an important step in this direction by showing that SiT encoding of fMRI spatial-autocorrelations can allow for CLIP decoding of *new* movie scenes, from subjects that the model was not trained on. Moreover, vision transformers are inherently interpretable allowing us to visualise the patterns of self-attention encoded by the model. Our results suggest that each attention head may be modelling different visual and semantic concepts. Thus far, this has only been assessed at a global level by comparing average patterns of attention against state-of-the-art models of functional organisation including gradient maps (Margulies et al., 2016), functional connectivity networks (Yeo et al., 2009) and the HCP multimodal parcellation (Glasser et al., 2016a). We are yet to explore whether self-attention varies meaningfully across individuals, including whether maps are predictive of behavioural or cognitive traits (Finn & Bandettini, 2021).

One notable current limitation with the current SiT architecture is how its computations scale with resolution. Increasing sampling of cortical patching by a single resolution level (from $I_3$ to $I_4$) increases the complexity of self-attention operations $16\times$. As this trade-off is critical for contrastive learning, exploring lighter forms of attention computation could help with scaling up the model. Another important consideration is the current lack of any model of sub-cortical fMRI, despite the known involvement of deep grey structures such as the Lateral Geniculate Nucleus (LGN) in vision (Ghodrati et al., 2017). In future this could be addressed by adding tokens for sub-cortical structures. Similarly, our study focuses on matching modalities based solely on $3s$ movie clips. While this represents an attempt at modelling temporal dynamics, increasing the length of samples might allow for modelling of more complex brain processes, such as memory and attention.

In doing so it may be important to consider expanding the dataset to include a wider range of audio-visual stimuli. While the HCP 7T movie-watching dataset is an incredible resource, collected across a large number of subjects and crucial to evaluate generalisation properties, it comprises a restricted set of audio-visual stimuli, collected from very different styles of movies. This constrains the amount of information that can be extracted and learnt by our model, meaning that we were unable to effectively test whether the model would generalise to completely different movies. In particular, this can also limits the potential for stimuli reconstruction (Figure 5) as the variety of visual signals is limited. Alternative datasets, which are richer in the amount of data collected per subject, include the 'Friends' fMRI dataset (Boyle J.A., 2020) and the 'Narratives' dataset (Nastase et al., 2021).

When considering the broader impact of such technologies, our interest is in tailoring brain-computer interfaces and neurofeedback therapies to individual brains, in which the needs and abilities of individual patients must be taken into account. Under these conditions, development of decoding models that generalise across brains are of vital importance, as they would allow for the sampling different experimental conditions across participants, requiring far less data collection for each new individual, while allowing models to simulate outside their training regime i.e. respond to novel situations. While, practical deployment of such healthcare models remains a long way off, it is important to be mindful of longterm potential ethical impacts of models that might generalise the decoding of human thought from brain activations; not least because all freely available open datasets are derived from healthy (largely Caucasian) controls, which risk model bias and poor generalisation to patient and minority groups.

ACKNOWLEDGMENTS

Simon Dahan would like to acknowledge funding from the EPSRC Centre for Doctoral Training in Smart Medical Imaging (EP/S022104/1). This project also made use of time on Tier 2 HPC facilities JADE2, funded by EPSRC (EP/P020275/1).

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

# A    APPENDIX - DATA PROCESSING

## A.1    7T TASK-FMRI HCP DATA

Functional MRI data were downloaded from the *Movie Task fMRI 1.6mm/59k FIX-Denoised* package available for download at `https://db.humanconnectome.org/`. Cifti files were separated into left and right hemispheres; then resampled from native resolution (59292 vertices) to $I_6$ resolution (40962 vertices). This resampling is necessary to integrate with the SiT framework, which utilises regular icosahedral grids (e.g. $I_3$) to patch the input surface data (at $I_6$).

## A.2    VIDEO & AUDIO DATA PROCESSING

All fMRI sessions (MOVIE1-4) were divided into non-overlapping $3s$ *.mp4v* movie-clips using `opencv`. Audio files were extracted to **.wav** format at 16kHz from all movie clips with `torchaudio` library.

**Choice of temporal length of stimuli**    The duration of $3s$ for movie clips was chosen for two reasons. First, it corresponds to the average 'movie-shot' duration for most movies shown during the different fMRI sesions as detailed in Table 2. Second, a three-frame reconstruction yielded the best results during the vsMAE pre-training, which motivated the use of $3s$ movie clips (as $TR = 1$). Qualitative and quantitative assessments of frame reconstruction are provided in Appendix C.7.

Table 2: Average 'movie-shot' duration (as defined in PySceneDetect) for each movie in each fMRI session. We used PySceneDetect, to extract discting 'shots' from each movie. PySceneDetect is a standard scene detection library with a simple algorithm based on changes between a rolling window of frames in the colour space to find fast cuts (the rolling window is to avoid considering fast camera movements as a scene cut). We verified visually that the output scenes were coherent.

| HCP 7T fMRI session | Movie sample ID | Average 'movie-shot' duration |
|---|---|---|
| MOVIE1_CC1 | 1 | 3.82 |
| MOVIE1_CC1 | 2 | 1.71 |
| MOVIE1_CC1 | 3 | 3.0 |
| MOVIE1_CC1 | 4 | 2.42 |
| MOVIE2_HO1 | 1 | 4.28 |
| MOVIE2_HO1 | 2 | 2.94 |
| MOVIE2_HO1 | 3 | 3.56 |
| MOVIE3_CC2 | 1 | 2.98 |
| MOVIE3_CC2 | 2 | 2.80 |
| MOVIE3_CC2 | 3 | 2.25 |
| MOVIE3_CC2 | 4 | 3.77 |
| MOVIE4_HO2 | 1 | 7.03 |
| MOVIE4_HO2 | 2 | 6.73 |
| MOVIE4_HO2 | 3 | 4.89 |

# B    APPENDIX - METHODS

## B.1    VSMAE PRE-TRAINING - ARCHITECTURE DETAILS

In the Table B.2, we summarise the vsMAE architecture used for pre-training the SiT encoder ($\Phi_{enc}^{fMRI}$) on video-frame reconstruction. Both encoder ($\Phi_{enc}^{fMRI}$) and decoder ($\Phi_{dec}^{fMRI}$) are based on the *SiT-small* architectures (Dahan et al., 2022). A classification token [CLS] is appended and used alongside the token sequence during the pre-training. It is used for the visualisation of attention maps.

Table B.1: All *SiT* models preserve a hidden size of 64 per attention head. The entire vsMAE encoder-decoder pipeline has in total 32M parameters. Here, the parameter count includes the initial linear projection layer from $I_3$ patch resolution (45 vertices per patch) and 3 fMRI frames (3 channels) to the embedding dimension $D$, and the final projection layer to resample the sequence to $I_6$ resolution.

| Models | Layers | Heads | Hidden size $D$ | MLP size | Params. |
|---|---|---|---|---|---|
| $\Phi_{enc}^{fMRI}$ | 12 | 6 | 384 | 768 | 21.3M |
| $\Phi_{dec}^{fMRI}$ | 6 | 6 | 384 | 768 | 10.7M |

## B.2 vsMAE pre-training - Training details

Here, we summarise some essential training details for the vsMAE pre-training task.

**Masking ratio**  While the temporal redundancy of pixels in natural videos, allows for effective agnostic masking with high masking ratios (of up to 90%) (Feichtenhofer et al., 2022; He et al., 2021), fMRI activations are less structured and much more noisy. Following results in Appendix C.7, we use a masking ratio of $\rho = 50\%$ in all vsMAE pre-training.

**Masking strategy**  Similarly, the lack of structure and low temporal resolution in the spatio-temporal dynamics of brain activity is not suitable for agnostic masking strategy as in (Feichtenhofer et al., 2022). Therefore, here we employ a tube-masking strategy as in Tong et al. (2022); Wang et al. (2023), where consecutive frames are masked with the same mask.

**Temporal modelling**  Comparatively to Dahan et al. (2024), here we do not concatenate the sequences of tokens from successive frames into a single token sequence; however, we concatenate fMRI patches from successive frames along the channel dimension. This design choice is primarily due to the complexity of the self-attention operation with long sequences.

## B.3 Stimuli Processing and Encodings

For each $3s$ non-overlapping movie-clip, video-latent feature representations were extracted from a pre-trained VideoMAE model (Figure 1.E) [3] (trained on the Kinetics dataset for video understanding (Kay et al., 2017)). Matched audio representations (Figure 1.F) were extracted using the wav2vec2.0 model (Baevski et al., 2020) pre-trained on 960 hours of unlabeled audio from LibriSpeech (Panayotov et al., 2015) and available via torchaudio (`torchaudio.pipelines.WAV2VEC2_ASR_BASE_960H`). Video and audio latent representations correspond to the token sequences extracted via *PyTorch hooks* from the penultimate layer of each network (i.e. prior to the classification head). In both cases pre-trained video and audio models were kept frozen during all trainings and experiments. Code for scene extraction can be found in: PySceneDetect [4].

## B.4 Framework - Architecture details and training time

In Table B.2, we report the number of parameters used by each component of the **SIM** pipeline.

Table B.2: Parameter counts for the different component of the **SIM** architecture.

| Models | Number of Parameters |
|---|---|
| SiT ($\Phi_{enc}^{fMRI}$) | 21.3M |
| Mapper $f_\theta^{fMRI}$ | 0.27M |
| Mapper $f_\theta^{video}$ | 0.44M |
| Mapper $f_\theta^{audio}$ | 0.44M |

---

[3] https://github.com/open-mmlab/mmaction2.
[4] https://github.com/Breakthrough/PySceneDetect/

The total training time for training the **SIM** pipeline (finetuning of the $\Phi_{enc}^{fMRI}$ and training of the multimodal mappers for tri-modal CLIP alignment) is of 24 hours on a cluster of 4 NVIDIA V100 GPUs with 32GB memory (internal cluster). Preliminary experiments (for development and hyperparameter tuning) took around 10 days of computing time on that same cluster. The vsMAE self-supervised pre-training took a total training time of 2 days.

## C  APPENDIX - RESULTS

### C.1  EXPERIMENTAL DESIGN

To further clarify the experimental design, we complement Figure 2 with an illustration of the multilevel train/test sampling setup in Figure C.1.

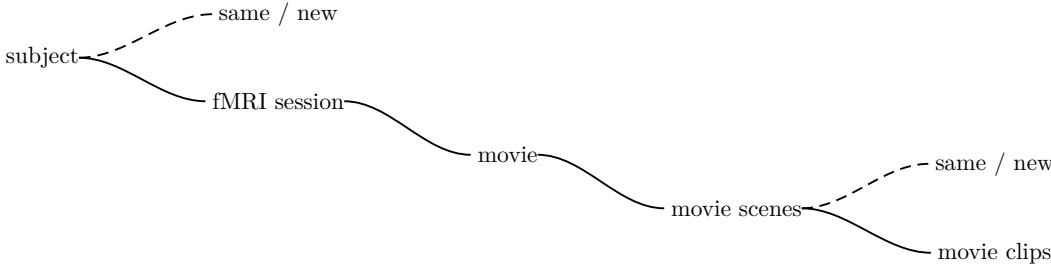

Figure C.1: Our multilevel sampling setup: the dashed lines highlight levels at which train and test time sampling diverges. For the remainder levels, no difference in is made train and test time.

### C.2  EXTENSIVE RETRIEVAL RESULTS - EXPERIMENTS 1,2 AND 3

Table C.1 is a more comprehensive version of Table 1 with results for *soft-negative* and *hard-negative* sampling in *Experiment 1*, including a sensitivity study to the training method (ablation study). Note, since the CLIP optimisation is performed symmetrically for any pairs of modalities, the evaluation pipeline can be similarly applied to encode fMRI activations from movie embeddings. We denote this evaluation as $\mathcal{V} \to \text{f}_{\text{MRI}}$ and $\mathcal{A} \to \text{f}_{\text{MRI}}$; showing that we are equally well able to predict fMRI from video or audio, as in reverse (Table 1 - bottom 2 rows). Table C.1 reports the results in four inference directions: $\text{f}_{\text{MRI}} \to \mathcal{V}$, $\text{f}_{\text{MRI}} \to \mathcal{A}$, $\mathcal{A} \to \text{f}_{\text{MRI}}$ and $\mathcal{V} \to \text{f}_{\text{MRI}}$. Results show that for all inference directions, performance drastically improves when we pre-train the SiT fMRI encoder ($\Phi_{enc}^{fMRI}$) on the vsMAE self-supervision task. This highlights the contributions of using surface reconstruction as a pre-training to learn spatio-temporal dynamics of brain activity. Fine-tuning further boosts top-1 performance for all models (and top-10 performance for most).

Table C.2 and C.3 compile all the results for *Experiment 2* and *Experiment 3* presented in Figure C.2 and Figure C.3.

Table C.1: Extensive results for the *Experiment 1*. We report top-1 and top-10 retrieval results when generalising to *new* subjects. **Bold** corresponds to the best-performing model for each inference modality. We compare random, ridge regression baseline and SiT encoder ($\Phi_{enc}^{fMRI}$) with our proposed tri-modal CLIP loss. We add here different combinations of training modalities, use of vsMAE pre-training (versus training from scratch), and fine-tuning. *training mode* refers to (1) frozen (training only the multimodal mappers (with 96% decrease in parameter count); (2) training from scratch; (3) finetuning from pre-trained weights. Results (in %) with $\bar{\mu}$ and 95% conf. interval (CI).

| Inference Modality | Methods | Training modalities | Pre-training | Training mode | *soft-negative* sampling top-1 | top-10 | *hard-negative* sampling top-1 | top-10 |
|---|---|---|---|---|---|---|---|---|
| $f_{MRI} \rightarrow \mathcal{V}$ | Random | ✗ | ✗ | ✗ | 2.0±0.3 | 17.5±0.6 | 3.7±0.5 | 19.4±0.9 |
| | Ridge | $f_{MRI}, \mathcal{V}$ | ✗ | (2) | 10.8±0.7 | 44.5±1.4 | 15.6 ±1.1 | 64.1 ±1.6 |
| | SiT | $f_{MRI}, \mathcal{V}$ | *vsMAE* | (1) | 67.7±2.3 | 92.4±1.7 | 57.6±2.2 | 94.4±1.5 |
| | SiT | $f_{MRI}, \mathcal{V}$ | ✗ | (2) | 39.2±2.5 | 87.7±2.6 | 50.2±2.5 | 93.5±1.7 |
| | SiT | $f_{MRI}, \mathcal{V}$ | *vsMAE* | (3) | 79.5±2.4 | **94.8±1.5** | 64.7± 2.3 | **95.6±1.4** |
| | SiT | $f_{MRI}, \mathcal{V}, \mathcal{A}$ | *vsMAE* | (1) | 42.0±1.7 | 83.2±1.9 | 37.6±1.7 | 89.7±1.6 |
| | SiT | $f_{MRI}, \mathcal{V}, \mathcal{A}$ | ✗ | (2) | 56.6±2.3 | 80.9±2.0 | 46.2±2.1 | 85.7±1.8 |
| | SiT | $f_{MRI}, \mathcal{V}, \mathcal{A}$ | *vsMAE* | (3) | **80.3±2.5** | 92.1±1.8 | **76.8±2.6** | 94.2±1.5 |
| $f_{MRI} \rightarrow \mathcal{A}$ | Random | ✗ | ✗ | ✗ | 2.9±0.4 | 30.9±1.2 | 3.1±0.4 | 30.1±1.1 |
| | Ridge | $f_{MRI}, \mathcal{A}$ | ✗ | (2) | 3.6±0.5 | 32.2±1.1 | 3.2±0.4 | 30.5±1.2 |
| | SiT | $f_{MRI}, \mathcal{A}$ | *vsMAE* | (3) | 24.1±1.4 | 64.1±1.7 | 19.9±1.3 | 62.5±1.7 |
| | SiT | $f_{MRI}, \mathcal{V}, \mathcal{A}$ | *vsMAE* | (3) | **70.9±2.8** | **87.9±1.9** | **56.6±2.5** | **86.9±2.0** |
| $\mathcal{V} \rightarrow f_{MRI}$ | SiT | $f_{MRI}, \mathcal{V}, \mathcal{A}$ | *vsMAE* | (3) | 74.8±2.9 | 90.7±2.1 | 52.5±2.5 | 89.8±1.8 |
| $\mathcal{A} \rightarrow f_{MRI}$ | SiT | $f_{MRI}, \mathcal{V}, \mathcal{A}$ | *vsMAE* | (3) | 61.1±2.8 | 88.0±2.0 | 50.6±2.6 | 85.0±2.1 |

Table C.2: Extensive results for the *Experiment 2*: Generalisation to *new movie scenes* for all subjects. Retrieval accuracy on new stimuli of training subjects. $M = 32$ is used to account for the size of left-out movie scenes. Results (in %) with $\bar{\mu}$ and 95% conf. interval (CI). The *hard-negative* results are shown in Figure C.3 and *soft-negative* results are shown in Figure C.2.

| Methods | Training Modalities | *soft-negative* sampling top-1 | top-5 | top-10 | *hard-negative* sampling top-1 | top-5 | top-10 |
|---|---|---|---|---|---|---|---|
| Random | ✗ | 3.6±0.2 | 17.9±0.4 | 33.2±0.5 | 4.6±0.3 | 20.4±0.5 | 39.8±0.6 |
| Ridge | $f_{MRI}, \mathcal{V}$ | 4.6± 0.3 | 19.1±0.5 | 35.1±0.6 | 7.1±0.4 | 30.0±0.7 | 58.2±0.8 |
| SiT | $f_{MRI}, \mathcal{V}, \mathcal{A}$ | **19.6 ± 0.7** | **53.8 ± 0.9** | **79.1± 0.7** | **13.9±0.5** | **43.3±0.8** | **70.2±0.7** |

Table C.3: Extensive results for the *Experiment 3*: Generalization to *new scenes* and *new subjects*. Retrieval Accuracy for new stimuli of testing subjects. $M = 32$ for inter-movie sampling but $M = 16$ for intra-movie sampling due to the limited size of left-out movie scenes. Results (in %) with $\bar{\mu}$ and 95% conf. interval (CI). The *hard-negative* results are shown in Figure C.3 and *soft-negative* results are shown in Figure C.2.

| Methods | Training Modalities | *soft-negative* sampling top-1 | top-5 | top-10 | *hard-negative* sampling top-1 | top-5 | top-10 |
|---|---|---|---|---|---|---|---|
| Random | ✗ | 4.5±0.4 | 17.4±0.7 | 30.5±0.8 | 5.5±0.4 | 21.1±0.7 | 40.0±0.8 |
| Ridge | $f_{MRI}, \mathcal{V}$ | 5.2±0.4 | 20.1±0.8 | 36.2±0.9 | 9.6±0.7 | 35.2±1.2 | 63.9±1.2 |
| SiT | $f_{MRI}, \mathcal{V}, \mathcal{A}$ | **19.2±1.1** | **50.1±1.4** | **79.2±1.1** | **12.8±0.9** | **43.6±1.2** | **80.7±1.1** |

## C.3 ATTENTION MAPS - COMPARISON WITH FUNCTIONAL NETWORKS

In Figure C.4, we compare the attention maps presented in Figure 4 against the three first gradients from Margulies et al. (2016) and the 7 resting-state networks from Yeo et al. (2011).

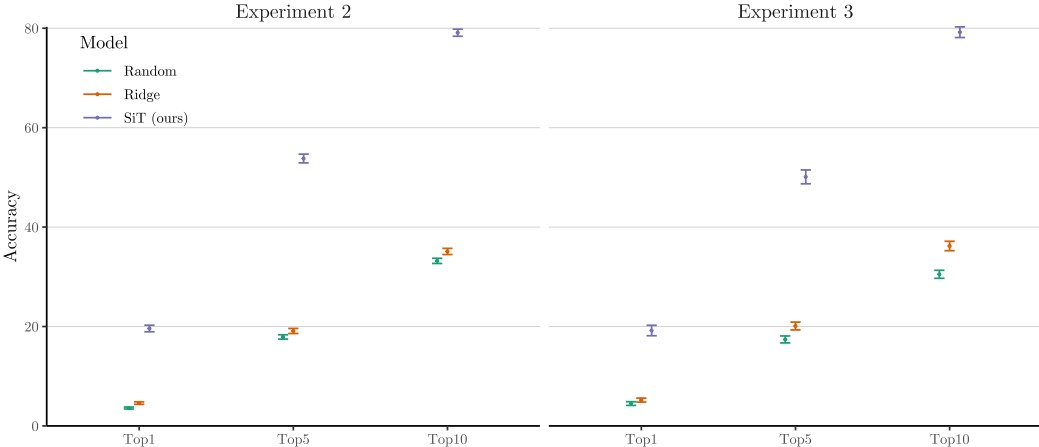

Figure C.2: *Soft-negative* $f_{\mathrm{MRI}} \to \mathcal{V}$ retrieval results for *Experiment 2* and *3*. Negative pairs (31) are sampled from different movies than the positive sample - but all sampled from *new movie scenes* (not used during training) - showing generalisation to new stimuli for train subjects (*Experiment 2*) and *new* subjects (*Experiment 3*), as detailed in Figure 2. Results (in %) with $\bar{\mu}$ and 95% conf. interval. Two-sample t-tests (Ridge VS SiT) with Bonferroni correction were highly significant ($p < 0.001$) .

Figure C.3: *Hard-negative* $f_{\mathrm{MRI}} \to \mathcal{V}$ retrieval results for *Experiment 2* and *Experiment 3*. Due to the experimental setup of *Experiment 3*, the number of samples is limited for *hard-negative* sampling. Therefore, here results are present with $M = 32$ for *Experiment 2* but $M = 16$ for *Experiment 3*. Results (in %) with $\bar{\mu}$ and 95% conf. interval. Two-sample t-tests (Ridge VS SiT) with Bonferroni correction were highly significant ($p < 0.001$) .

## C.4 ATTENTION MAPS ANALYSIS

Using the 50 topics from the Neurosynth maps, we correlated the attention maps shown in Figure 4 to highlight the brain function associated with each attention heads. We show the result in Figure C.6. Topic can be found `https://neurosynth.org/analyses/topics/v5-topics-50/`. This results confirm the specialisation of each attention heads to extract signal from specific functional networks.

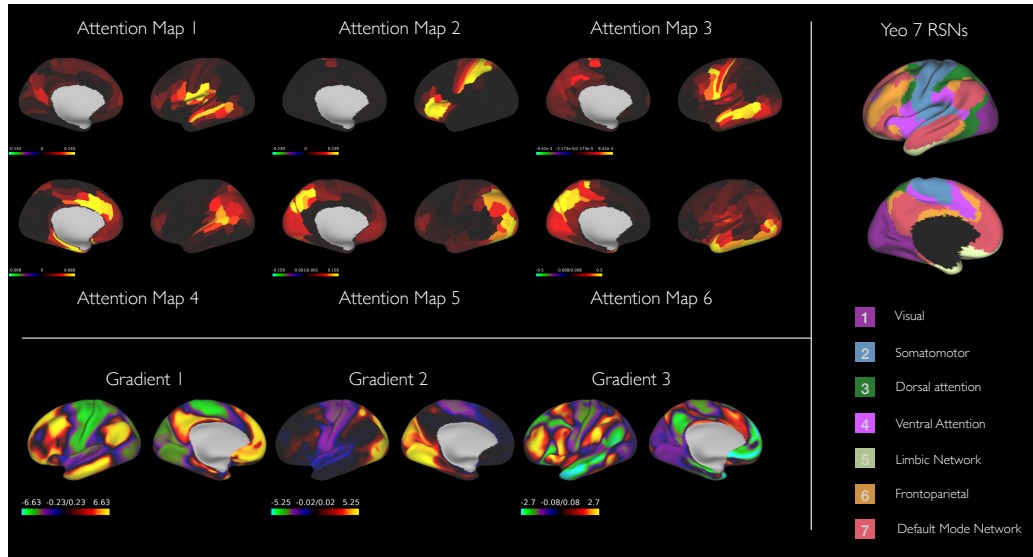

Figure C.4: Comparison between the attention maps extracted as in Figure 4, Margulies et al. (2016) gradients maps and Yeo et al. (2011) functional networks

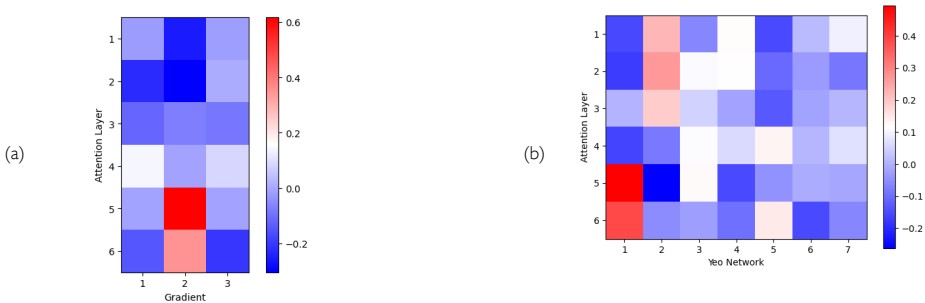

Figure C.5: Pearson correlation between Margulies et al. (2016) gradient maps, Yeo et al. (2011) functional networks and the six average attention heads as described in Figure 4.

## C.5 RETRIEVAL RESULTS

Additional decoding results are provided, while retrieving from $f_{MRI} \rightarrow \mathcal{V}$. Figures C.7 and Figure C.8 show retrieval results, respectively, for intra-movie sampling and inter-movie sampling for unseen subjects (trained on the same stimuli). These retrieval results provide additional insights into the generalisation capabilities of the tri-modal clip alignment for stimuli retrieval in a large population. Figure C.9 provides additional results for stimuli retrieval in *unseen subjects* and *unseen stimuli*.

Figure C.8 and Figure C.7 show visual examples for decoding performance when inferring $f_{MRI} \rightarrow \mathcal{V}$. This shows that not only is the selected top-1 clip often correct but also that all frames in the top 3 show semantically similar information. By contrast the bottom 3 clips are clearly very different.

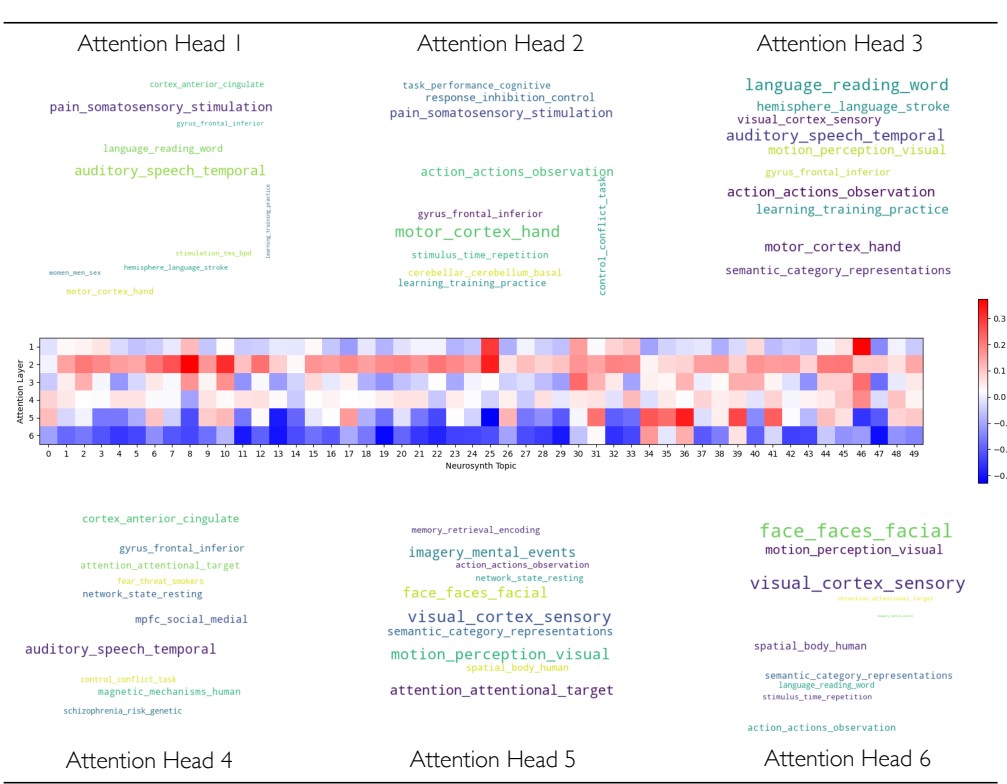

Figure C.6: Neurosynth topic correlation with Attention maps shown in Figure 4

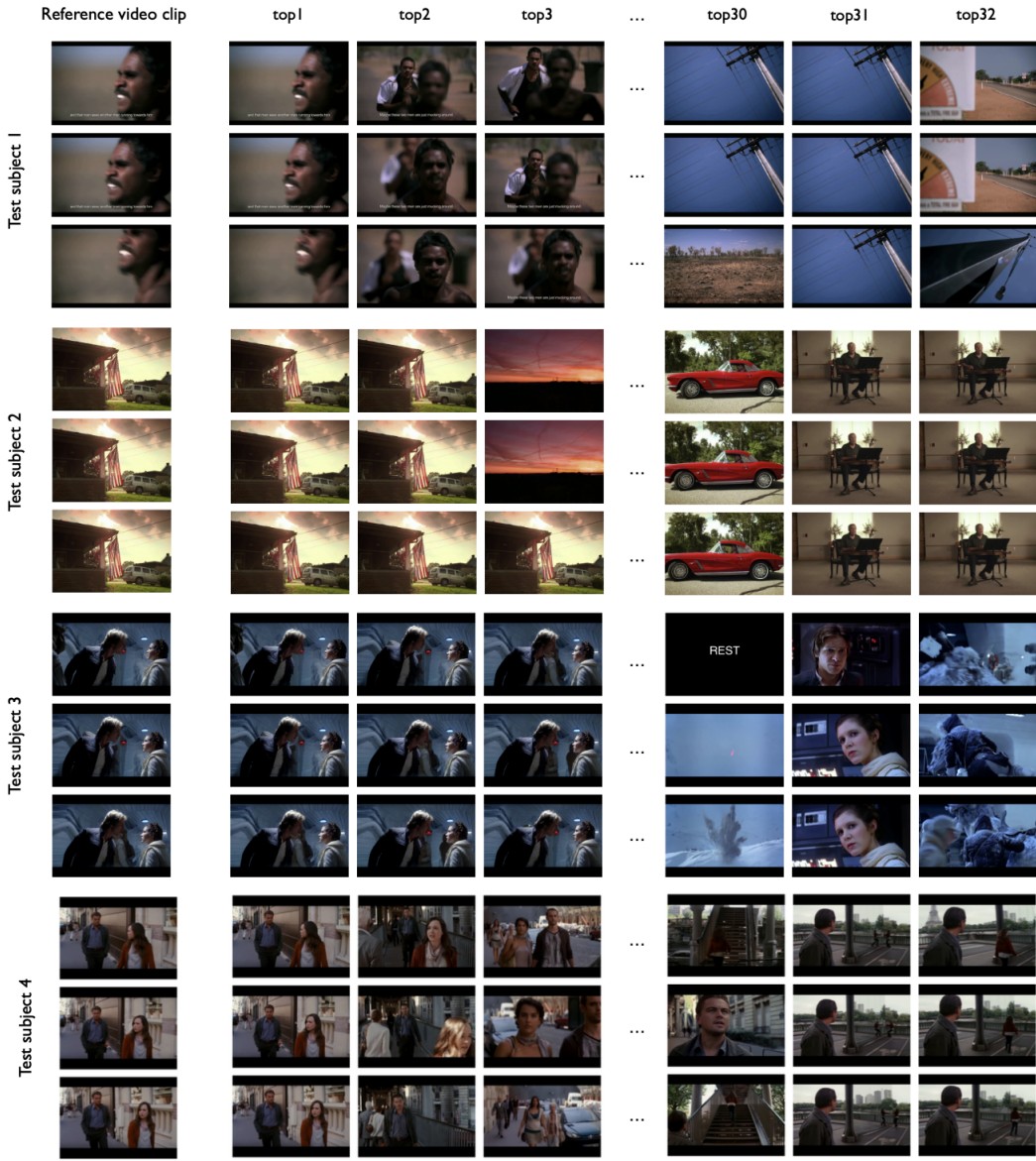

Figure C.7: *hard-negative* sampling results for *new* subjects. The model can generalise stimuli-retrieval for *new* subjects (trained on the same stimuli). The top-ranked movie-clips are semantically close.

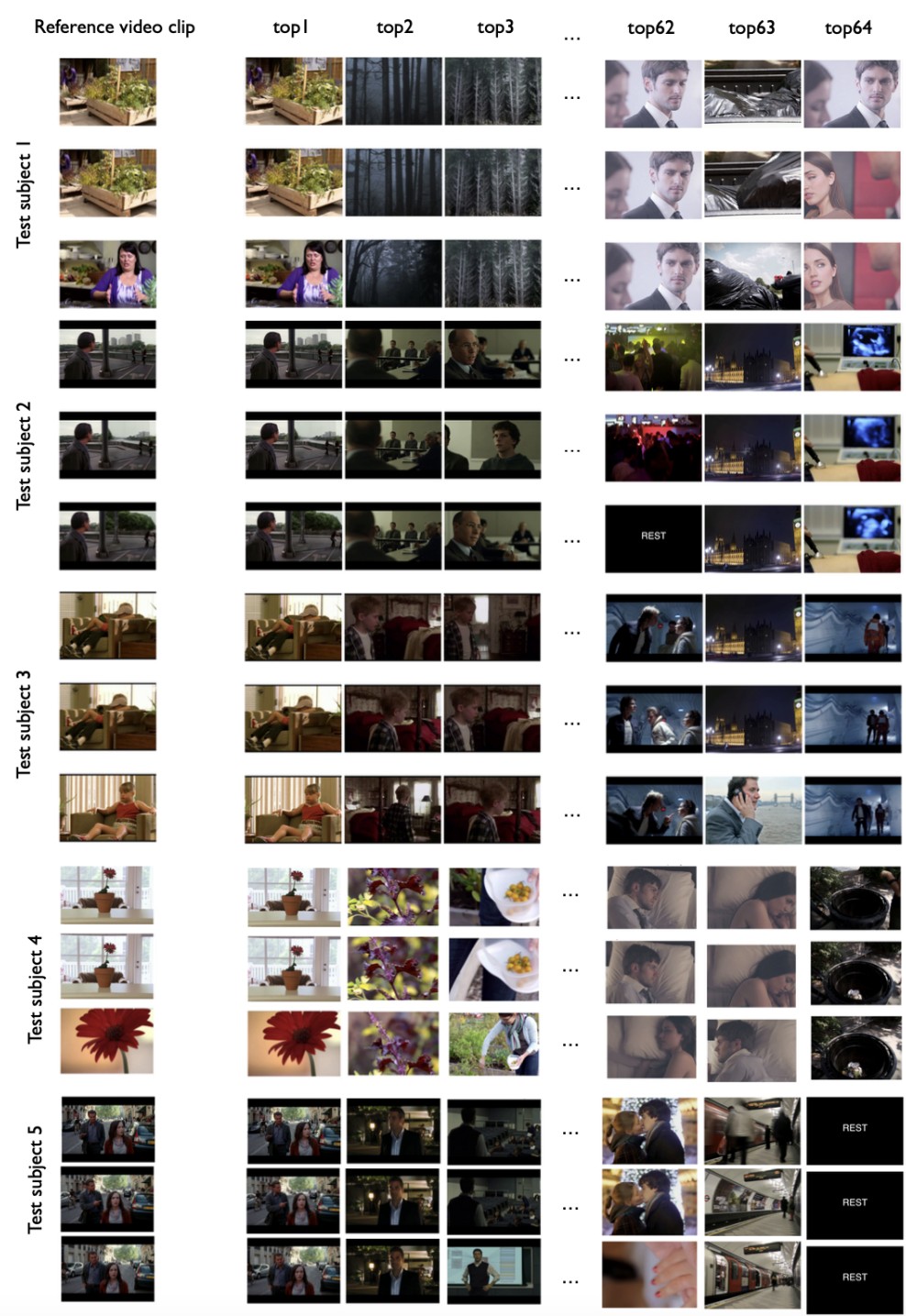

Figure C.8: *soft-negative* sampling results for *new* subjects. Here, the model can correctly rank stimuli among a large pool of movie-clip samples (64), clustering similar concepts together from different movies: nature (subject test 1 and 4), people talking (subject test 2 and 5), kids (subject 3).

Figure C.9: Video retrieval for *unseen scenes* and *unseen subjects*. (a) corresponds to inter-movie sampling: top-ranked movie clips are all dialogue scenes; (b) corresponds to intra-movie sampling: top-ranked video clips all show human and bright-color stimuli.

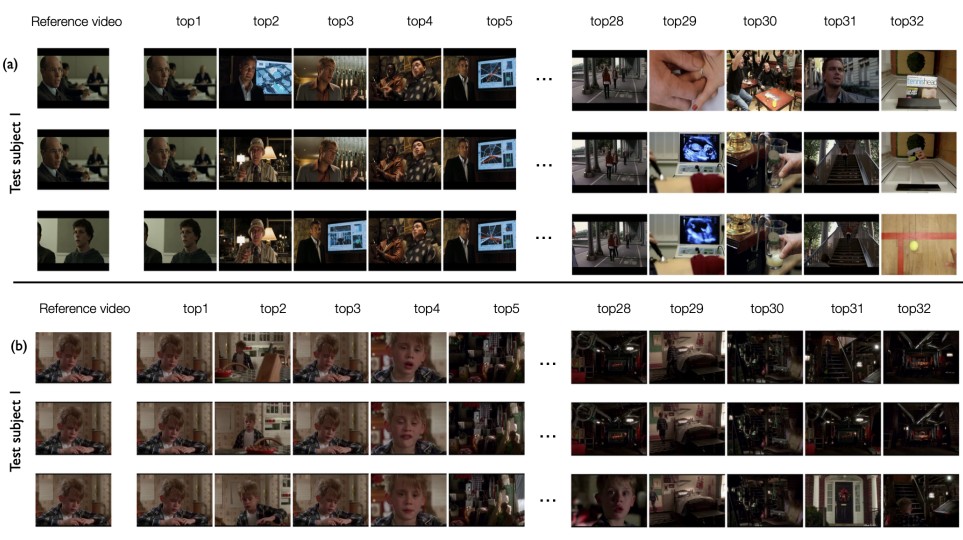

## C.6 RECONSTRUCTION RESULTS

In Figure C.10 and Figure C.10, we provide additional video-frame reconstruction results from the *Experiment 1*. Figure C.10 shows the variability in the reconstruction results between different test participants, while C.11 shows the diversity in semantic reconstruction. Overall, our **SIM** model can generalise to *new* subjects without having to fine-tune or train a model on test subjects.

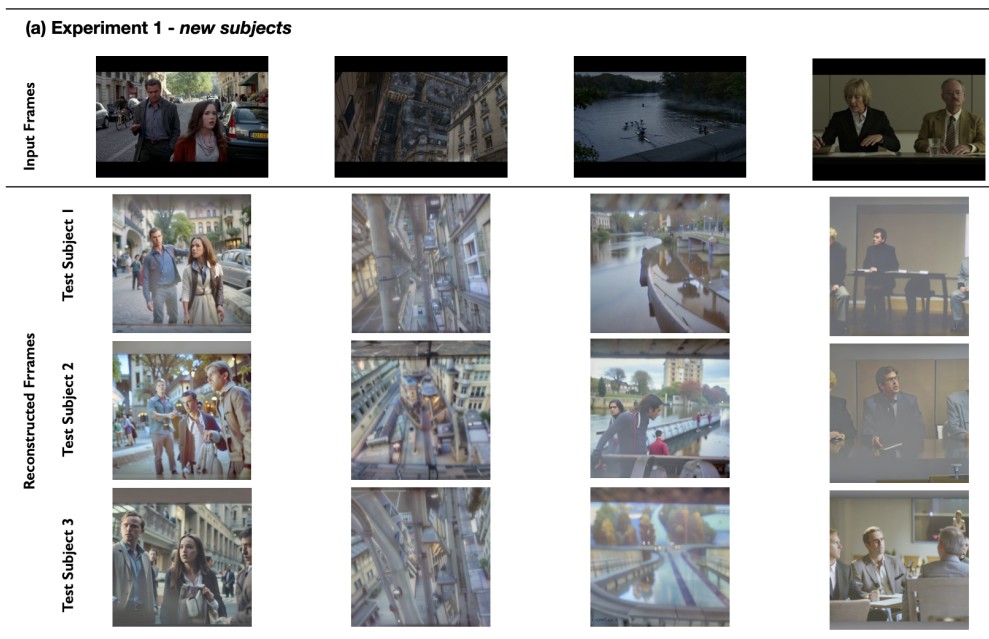

Figure C.10: Video-frame reconstruction results from fMRI embeddings after tri-modal CLIP alignment. Here, we show the reconstruction results for 3 test subjects for the same movie-frame, demonstrating that the **SIM** framework can generalise to individual brain activation.

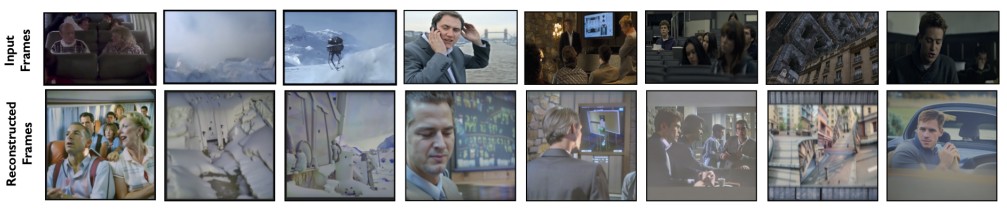

Figure C.11: Video-frame reconstruction results from fMRI embeddings after tri-modal CLIP alignment. Here, we show various video-frame reconstructions for one test subject, showing that the **SIM** framework captures various semantics.

## C.7 VMAE PRE-TRAINING - MASKING RATIO

The masking ratio parameter is a crucial hyperparameter in the vsMAE pre-training task as it impacts both reconstruction quality and training time. We evaluated the performance of various masking ratios (25%, 50%, 75% and 90%), on frame reconstructions for the task fMRI HCP data. In Table C.4, we report the average MSE video-frame reconstruction error over 3 training runs, for different masking ratios $\rho \in [25\%, 50\%, 75\%, 90\%]$.

| Masking Ratio $\rho$ | MSE Reconstruction Error |
|---|---|
| 25% | $0.78 \pm 0.05$ |
| 50% | $\mathbf{0.39} \pm 0.03$ |
| 75% | $0.49 \pm 0.03$ |
| 90% | $0.68 \pm 0.05$ |

Table C.4: Reconstruction errors (MSE) are reported on the validation set for the 3-frame reconstruction task. MSE is calculated only for the masked tokens. In all cases, vsMAE self-supervision was done for 50,000 iterations with a batch size of 64. Validation errors with stds across three runs are reported.

Qualitatively, we report reconstruction results in Figure C.12. The reconstruction quality is high, capturing dynamics of functional connectivity over successive frames, even with high-masking ratios.

Overall, the 50% masking ratio offered the best quantitative validation and was used in all experiments. Such observations are aligned with those in He et al. (2021), where it was noted that masking a high percentage of patches is necessary to reduce redundancy and create a challenging self-supervisory task which leads to learning more meaningful weights; however, increasing the masking ratio above a certain threshold leads to higher reconstruction errors and consequently higher prediction errors.

## C.8 TEMPORAL LAG

The choice of temporal lag of $\tau = 6$ was validated by calculating the correlation between the video latent embeddings ($\mathcal{X}_{\mathcal{V}}$) (extracted from the pre-trained videoMAE (Tong et al., 2022) and $3s$ movie-clips) and the corresponding $3s$ delayed by various temporal lags $\tau \in [1, 3, 6, 10]$, following methodology in Huth et al. (2016). This was achieved by training a ridge regression model for each HCP participant, on compiled MOVIE1-3 fMRI sessions and predicting the individual timeseries for the missing fMRI session (MOVIE4). An example of the map after statistical correlation Wilcoxon and Bonferroni correction, shows vertices with $p\text{-value} < 10e^{-8}$ in Figure C.15. By using a temporal lag of $\tau = 6$, the correlation values were the highest.

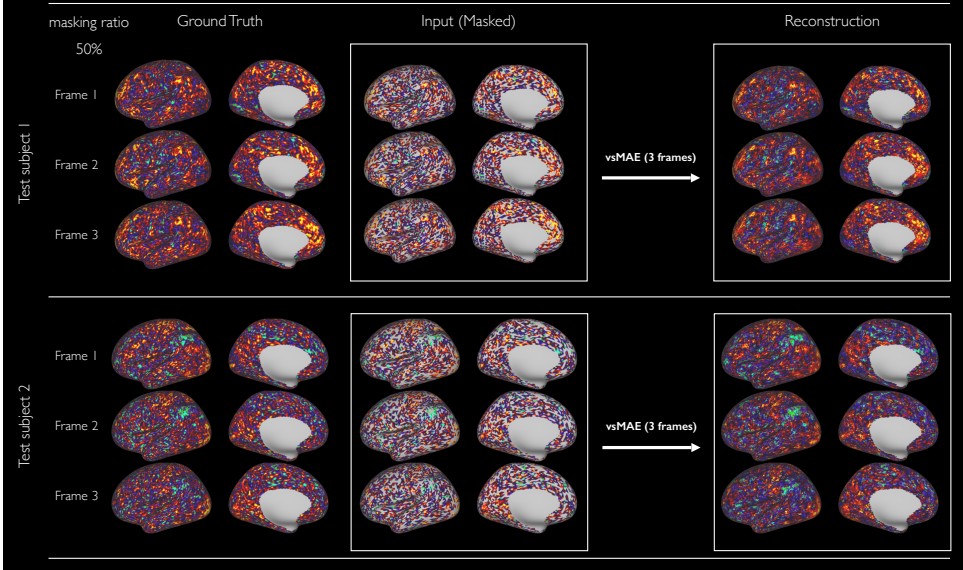

Figure C.12: fMRI reconstructions result from vsMAE pre-training on 3-frames reconstruction and trained with masking ratio $\rho = 50\%$. Results for two test subjects are presented, showing high-quality reconstruction of individual brain activation.

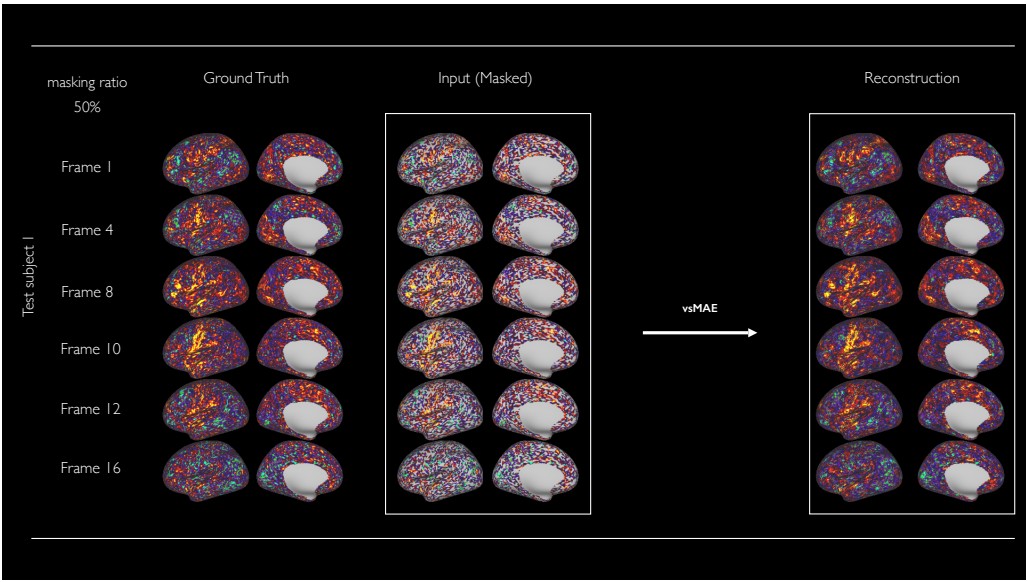

Figure C.13: fMRI reconstructions result from vsMAE pre-training on **16-frames** reconstruction and trained with masking ratio $\rho = 50\%$. Reconstruction results are presented for 6 frames from a 16s time window. The vsMAE effectively captures spatio-temporal dynamics of brain activity while preserving individual features.

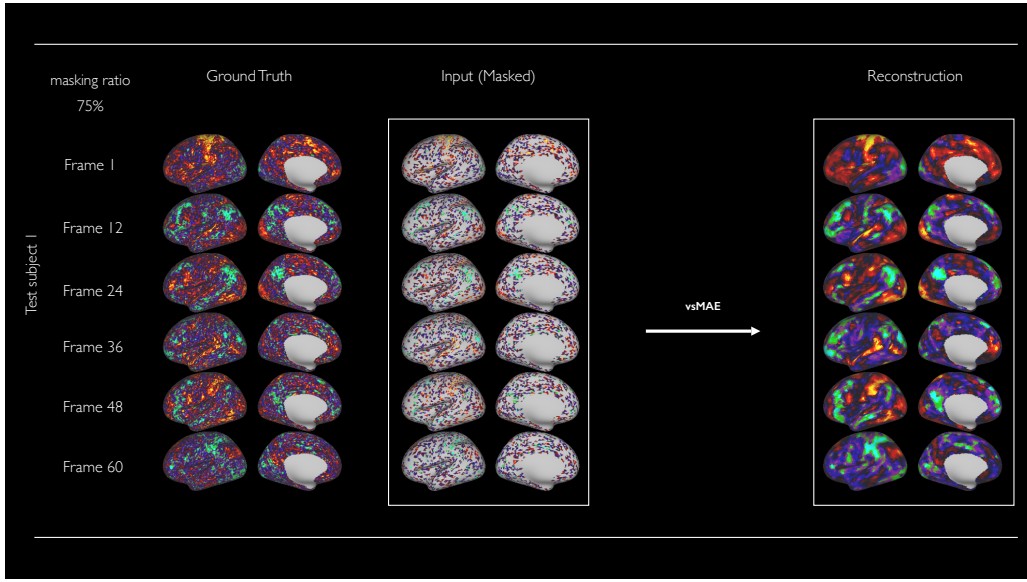

Figure C.14: fMRI reconstructions result from vsMAE pre-training on **64-frames** reconstruction and trained with masking ratio $\rho = 75\%$. Reconstruction results are presented for 6 frames from a 64s time window. The vsMAE captures global patterns of brain activity in the form of functional networks.

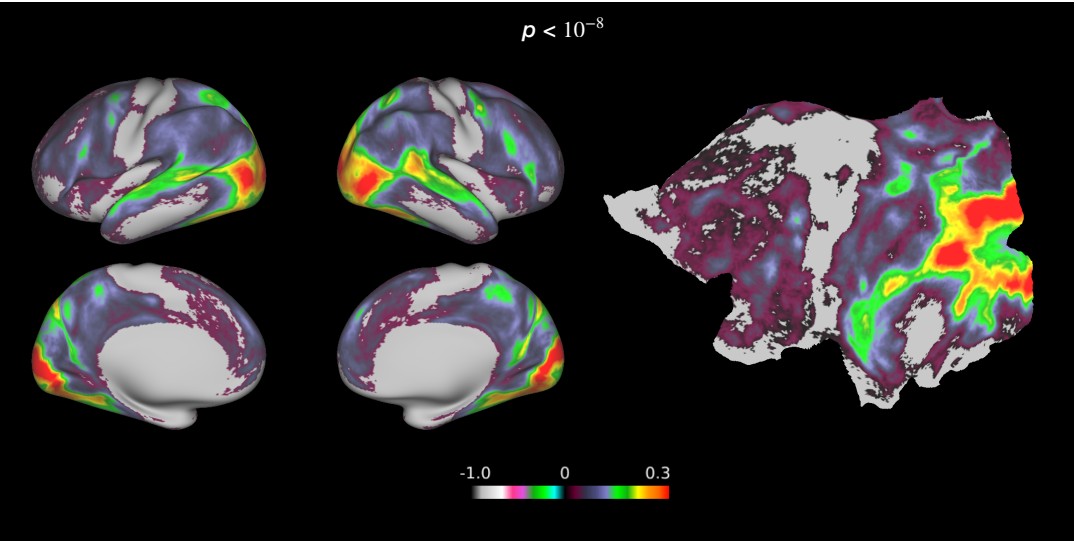

Figure C.15: Correlation map of predicted fMRI timeseries from video latent representations for the fMRI session MOVIE4, with a temporal lag of $\tau = 6$. These maps are averaged across test subjects and statistically corrected for Wilcoxon and Bonferroni correction. Most of the visual and auditory systems are highly correlated.

