# OpenReview forum: "SIM: Surface-based fMRI Analysis for Inter-Subject Multimodal Decoding from Movie-Watching Experiments"
_ICLR.cc/2025/Conference — ICLR 2025 Poster_

### Official Review · Reviewer_6xCn · 2024-10-15

**Soundness:** 3
**Presentation:** 2
**Contribution:** 3
**Rating:** 6
**Confidence:** 4

**Summary:**

This paper presents a novel approach for brain decoding using surface vision transformers (SiT) and tri-modal CLIP contrastive learning to generalize audiovisual stimuli decoding from fMRI data across different subjects and unseen stimuli. The model leverages the spatial topography of the brain’s cortical surface, combined with alignment between fMRI, audio, and video embeddings, achieving up to 80% top-1 accuracy in identifying stimuli across subjects in a movie-watching task. The approach is validated on the HCP 7T movie-watching dataset, showing strong generalization and offering interpretability through attention maps that highlight brain regions involved in audiovisual processing.

**Strengths:**

- The use of tri-modal CLIP alignment for fMRI, audio, and video embeddings is an innovative contribution, extending prior unimodal or bimodal approaches.

- The integration of state-of-the-art techniques, like vision transformers and contrastive learning, is well-grounded and technically solid. Also, the model is rigorously evaluated on unseen subjects and stimuli, demonstrating strong performance in terms of Top-K accuracy.

- Its potential impact on brain-computer interfaces (BCIs) and neuroscience research is significant, with promising future applications.

**Weaknesses:**

**Limited Generalization Beyond the Dataset**: The model’s generalization is limited to the HCP 7T dataset, with no evaluation on more diverse or real-world stimuli.

**Computation**: the model’s self-attention mechanisms are computationally expensive, limiting scalability to higher resolutions.


Some of the typos that need to be corrected:

1. Appendix C.1, under Table C.1.'s caption. "3-frame reconstruction taskon", should be "3-frame reconstruction task on"?
2. Line 512, "evalute" should be "evaluate"

**Questions:**

1. In section 4.4, the authors propose the potential for training encoding/decoding frameworks that exhibit generalizability. While the model shows robust generalization performance on the HCP 7T movie-watching dataset, have the authors considered evaluating its generalizability under more diverse experimental conditions, such as with abstract visual stimuli, silent films, or non-narrative content? Given that the HCP dataset primarily includes specific cinematic types, how do the authors anticipate the model’s performance on stimuli with varying visual and auditory properties?


2. In Table 1:
-  The vsMAE pre-training appears to significantly enhance performance compared to training from scratch. For instance, in the case of $f_{MRI}\rightarrow \mathcal{V}$ with SiT, vsMAE pre-training substantially improves the top-1 metrics. Could the authors elaborate on how this pre-training enhances the model’s generalization capabilities? Is this improvement consistent across different stimulus types (e.g., non-movie visual content), or is it specifically tied to the training dataset?

   - Some results exhibit considerable variance. For example, the results for $f_{MRI} \rightarrow \mathcal{A}$ with SiT show $70.9 \pm 20.3$, and $f_{MRI} \rightarrow \mathcal{V}$ shows $80.3 \pm 18.0$. Could the authors provide an explanation for this variability?

3. I am a little bit confused about the information that Figure 4 wants to present. Could the authors clarify how these maps are quantitatively validated against known functional networks (Yeo et al), and how confident we can be that these maps are genuinely reflecting underlying neural processes rather than model artifacts?

4. The overall generalizability of the model was evaluated using the same dataset. From my understanding, the contribution to generalizability here is that the model improves upon previous methods (which focused on single-subject analysis) by making it more generalizable to future tasks, but still within the same study population or dataset. Is this correct? Could the authors comment on this? Typically, when referring to generalizability, it implies extending the model’s applicability to a broader population or more diverse datasets.

5. In the discussion section, the author mentioned that the current model does not include sub-cortical structures, despite their known role in sensory processing, such as the Lateral Geniculate Nucleus (LGN) in vision. Is it viable to handle the noise in subcortical area using CIFTI dataset?

6. The authors utilize sine-cosine positional embeddings to encode the locations of cortical patches in the SiT model. Have the authors evaluated the effectiveness of this choice for representing the complex geometry of the brain’s surface? Would alternative methods, such as learned positional embeddings, improve performance or better capture spatial relationships in cortical data?

7. What's the computation time?

---

> ### Author Response · Authors · 2024-11-22
> **Rebuttal [1/3]**
>
> We want to thank the reviewer for their comments on the potential impact of this work and the robustness of evaluations. They raise some really important questions that we seek to address below:
>
> ### Weaknesses
>
> #### **1. The computational expense of  MHSA**
>
> In weaknesses, the reviewer questions the use of multi-head self-attention (MHSA) operations due to their computational complexity, which would limit their scalability to higher resolutions. However, this is mitigated through working on the surface instead of in the 3D volume. Our patches are extracted from the $I_3$ icosphere, which, when mapped to the native cortical surface, have a mean area of $52.25mm^2$ (per patch) - allowing fine-grained modelling; moreover, features are actually sampled from $I_6$ with resolution of $1.2 mm^2$, which is higher resolution than the acquired fMRI volume (1.6mm isotropic).
>
> While the computational cost of self-attention is a well-known limitation of transformer models, the results from the SiT paper (Dahan 2022) suggest that this is a worthwhile trade-off given the challenges in translating convolutions to non-Euclidean domains (such as surfaces), which lack a global coordinate system on which to constrain the convolutional filtering operation. This was demonstrated through benchmarking the SiT and surface CNNs on a range of developmental phenotype regression tasks.
>
> #### **2. The sole use of the HCP 7T dataset**
>
> Our decision to utilise the HCP 7T movie-watching dataset was driven by an objective to demonstrate generalisation across subjects. This dataset provides a substantial number of examples from diverse brains, enabling us to effectively model individual cortical organisational variability in a generalisable manner. Previous studies, such as Glasser et al 2016. (Nature), have demonstrated that the HCP dataset effectively captures significant individual variability, thereby supporting our approach. The HCP 7T fMRI movie-watching dataset scanned subjects 4 times, for 15 mins each; 174 participants were shown short (1-4.3mins) but diverse clips of different movies. This should be put into perspective of other datasets used for decoding e.g. the NSD dataset (used by Ozcelik 2023; Scotti 2023; 2024; Thual 2023), which is densely sampled but acquired in a very small number of subjects - eight individuals with 8859 static images, shown 3x each over 40 scanning sessions. These papers typically train and test models within an individual brain; if they do show generalisation e.g. Thual 2023 and Scotti 2024 they need at least an hour of data for the test subject (to perform an alignment step); whereas we use just 3 seconds.
>
> #### **3. Typo**
>
> We apologise for the typos and hopefully have corrected all of them. Thank you for pointing them out.
>
> ### Questions
>
> #### **Q1. Testing on more diverse experimental conditions**
>
> We agree with the reviewer that evaluating our model on a more diverse range of datasets with atypical stimuli, such as abstract visual stimuli, silent films and non-narrative content, would provide valuable insights. However, the HCP 7T movie-watching experiment was meticulously designed to be as diverse as possible, and this includes a range of blockbuster and independent movies with some clips containing little or no dialogue, or blockbusters, where the movie scenes were carefully selected based on various criteria (scene dynamics, motion changes, luminance and colour contrast), making them highly diverse while relatively short `[1]`. For instance, some scenes feature long dialogue with minimal motion, while others involve intense action with little dialogue. For these reasons, we believe that the capabilities of our model to retrieve and reconstruct such diverse stimuli just should indicate strongly that the model would generalise to other types of (more densely sampled) complex stimuli.

---

> ### Author Response · Authors · 2024-11-22
> **Rebuttal [2/3]**
>
> #### **Q2. Impact of vsMAE pre-training**
>
> From a biological modelling standpoint, the purpose of the vsMAE (SiT encoder) is to build a transformation-invariant model of the spatial topography of cortical organisation (placement and shape of functional activations) and its temporal dynamics. By randomly masking the cortical surface in space and time, the pretraining of the vsMAE teaches the model to attend to functional networks such that it can then recognise them in the decoding task. New figures of fMRI reconstructions are provided in Figures C.11, C.12 and C.13, highlighting that the vsMAE pre-training allows the SiT encoder to learn individual features of brain activation.
>
> This approach was also validated through visual QC of the attention maps (main manuscript Figure 4), which clearly highlight known sensorimotor, auditory and visual networks. We have now annotated Figure 4, showing the brain areas implicated in movie-watching. The features learned during this pretraining phase not only facilitate faster convergence of the final model but also significantly enhance overall performance in all inference directions (see full results and ablation study in Table C.1). Notably, models trained from scratch require substantially longer training times to achieve comparable results, demonstrating that vsMAE pre-training provides a crucial advantage in improving both the efficiency and accuracy of stimulus decoding.
>
> **Addressing variability** Given the variability of the stimuli and the relatively sparse sampling of the HCP 7T fMRI movie-watching dataset some clips are far more difficult to decode than others. We expect the variance to be significantly reduced on datasets with denser sampled stimuli and/or we could consider looking at the attention maps of poor-performing clips to see if we can provide engineering solutions to these problems.
>
> #### **Q3. Validation of the attention maps**
>
> The attention maps were quantitatively validated and qualitatively compared against the Yeo networks and Margulies Gradient maps by using MSM surface registration (Robinson 2018)  to map them all into the same surface template space in which they overlap. We assessed similarity using Pearson’s correlation, which, for functional networks (Yeo - 7 RSNs), reduces to estimating the overlap between RSNs and areas the SiT encoder most attends to. For the gradient maps, the correlation is comparing the overall patterns of the cortical gradient maps and the attention maps. Following the reviewer's recommendations and to enhance clarity, we have added visualisations of these comparisons to Figures C.2 and C.3 of the appendix and annotated the attention maps in Figure 4.
>
> The attention maps show the cortical regions that the model prioritises when learning low-dimensional CLIP representations that align the fMRI data with audio and video embeddings of movie scenes. To achieve this, the model has to learn a highly compressed representation by focusing only on the most relevant signal for predicting each scene. This indicates that the attended areas correspond to regions known to be important for movie-watching. For stimuli that are less well-understood, such attention maps could provide insights into the underlying neural processes involved in encoding these stimuli.
>
> #### **Q4. Defining what we mean by generalisation**
>
> In this paper, we define generalisation as *the model’s ability to accurately decode stimuli from subjects that were not included during training*. This represents an improvement over recent decoding and reconstruction studies, which typically train and test within data collected from the same individual. Similar terminology is utilised by Scotti et al. (2024) and Thual et al. (2024). We have also created a new illustration in Figure 2 to simplify the description of the experimental setup and the intended generalisation property of each experiment.
>
> While our current validation focuses on generalising within the HCP 7T movie-watching dataset, we do not preclude the possibility of extending our approach to other datasets. In future work, we aim to investigate whether our model can predict an individual’s brain responses to various tasks, potentially using only their resting-state data. Generalising across more different datasets would likely require an alignment step to account for variations in spatial and temporal resolution. We agree with the reviewer that this represents an important and intriguing direction for future research, and we believe that our strong performance on the challenging HCP 7T dataset provides a solid foundation for such extensions.
>
> #### **Q5. Addressing noise in subcortical areas**
>
> Our understanding is that noise should be improved in HCP 7T (relative to 3T). We are yet to try this but visualising the attention maps would tell us whether it is useful or not.

---

> > ### Author Response · Authors · 2024-11-22
> > **Rebuttal [3/3]**
> >
> > #### **Q6. Positional embeddings**
> >
> > Thank you for raising this important question. Positional embeddings provide a crucial inductive bias for modelling the spatial relationships between cortical surface patches. In the vsMAE paper (Dahan et al. 2024), it was demonstrated that sine-cosine positional embeddings outperform learned positional embeddings in reconstruction tasks. This advantage arises because sine-cosine embeddings supply explicit positional information, whereas learned embeddings are initially random. Consequently, models utilising sine-cosine embeddings can more effectively and rapidly use the spatial organisation of patches, leading to faster convergence for the reconstruction task in vsMAE. Therefore, we opted to employ sine-cosine embeddings for both the pre-training reconstruction task and the subsequent multimodal CLIP self-supervision.
> >
> >
> > #### **Q7. Computation time**
> >
> > Appendix B. 4 details the computation time for the two training phases (vsMAE and CLIP). In short, training the CLIP model, which included fine-tuning the vsMAE encoder and training the multimodal mappers from scratch, all together and with the CLIP loss, took about 24 hours (in the setting that we used). For the vsMAE pre-training task, we empirically found out that the longer the training, the better the results are. However, the vsMAE pre-training was stopped after 2 days, as the results were already satisfactory. For the CLIP training, we used all the resources available as it has been shown that the batch size impacts the alignment between modalities, as there are more negative pairs to contrast with the positive sample. However, it is not essential to train on multiple GPUs, and the code released upon publication will work for single or multiple GPU training. Other fMRI decoding studies use similar resources ( Defossez et al 2023, Benchetrit et al 2023 etc.) but with fewer subjects and smaller spatial resolutions (either voxels extracted from the cortical surface or MEG/EGG). In some extreme cases (MindEye2), the amount of recorded information is massive (8 subjects with 30 to 40 hours of recording per subject) which requires larger computational resources (8 A100 GPUs).
> >
> > ---
> >
> > 1. Cutting, J. E., Brunick, K. L., & Candan, A. (2012). Perceiving event dynamics and parsing Hollywood films. Journal of Experimental Psychology: Human Perception and Performance, 38(6), 1476–1490. https://doi.org/10.1037/a0027737
> >
> >
> > ---
> > ---
> > ---

---

> > > ### Comment · Reviewer_6xCn · 2024-11-23
> > > **Changing my rating from 5 to 6**
> > >
> > > I'm satisfied with the response from the authors, thereby I increase my rating from 5 to 6.

---

### Official Review · Reviewer_vWQL · 2024-10-24

**Soundness:** 3
**Presentation:** 3
**Contribution:** 2
**Rating:** 6
**Confidence:** 3

**Summary:**

The paper addresses the challenge of generalizing decoding models for video and audio stimuli to fMRI responses from subjects not included in the training data, aiming for a universal cognitive model rather than subject-specific ones typically achieved in this field. The authors propose an approach that combines anatomical and functional alignment to map across subject-specific response spaces. Their method leverages two main principles: topographical alignment and space-time pattern matching. To implement this, they use a Vision Transformer with cortical surface response patches as input, alongside a tri-modal CLIP criterion imposed on a shared embedding space for audio, video, and fMRI. The approach is evaluated on the 7T HCP movie dataset with over 170 subjects, demonstrating successful nearest-neighbor identification of video and audio samples from responses of held-out subjects. Additionally, some success is shown in the fully zero-shot setting, where both the test subject and test stimuli are unseen during training.

**Strengths:**

- Significant Challenge Addressed: The paper tackles the important and well-recognized challenge of inter-subject generalization in brain decoding tasks. By striving for a universal cognitive model that can apply to responses from unseen subjects, the authors make a meaningful contribution to the field, which often relies on subject-specific models that limit broader applicability.

- Innovative Methodology: The authors present an intriguing combination of a Vision Transformer (ViT) applied to patches of cortical surface responses, paired with a multi-modal contrastive learning criterion. This approach leverages the strengths of ViT for spatial representation while effectively integrating different modalities (audio, video, fMRI) to enhance the decoding process.

**Weaknesses:**

The main issue is the combination of too many ideas within a single paper. While the focus on inter-subject generalization in stimulus decoding is critical, the inclusion of multi-modal audio and video adds complexity that obscures the primary contributions. The rationale for combining these modalities should be clearly justified, particularly since generalization across subjects poses challenges even in simpler stimulus domains like images. A more focused exploration of the decoding problem could strengthen the paper's impact.

The results are difficult to interpret, especially in Table 1. The meaning of the "inter" and "intra movie" columns needs clarification. It would be beneficial to restrict the table to test movies included in the training dataset. Additionally, an evaluation of audio versus video identification success is lacking. The absence of same-subject (intra-subject) baseline results and chance-level metrics makes it challenging to assess the significance of the reported accuracies (e.g., 80% vs. 77% top-1 accuracy). The authors should enhance guidance in the text to help the reader grasp the key contrasts of interest.

Furthermore, the decoding problem addressed appears to be a nearest-neighbor classification, which is arguably a basic form of decoding. Given the recent advancements in within-subject stimulus reconstruction, a more compelling approach would explore inter-subject generalization in the context of these more popular and challenging brain decoding tasks.

Additional comments:
- The originality of the work is challenging to evaluate. The authors should more thoroughly situate their contributions within the context of previous research, particularly in relation to cited works by Dahan et al. (2024), Tong et al. (2022), and Feichtenhofer et al. (2022), emphasizing the added value of their approach.
- The paper lacks plots that could effectively convey findings in a more visual and intelligible manner. Incorporating proper uncertainty estimates in graphical representations would enhance clarity.
- The writing style is overly cryptic, with excessive notation making the paper difficult to follow. Simplifying the presentation could greatly improve readability.
- The figures, particularly in the main figure (Fig. 1), suffer from poor quality; the font sizes are too small, and the caption is not sufficiently explanatory.
- There are numerous grammatical and syntactical errors throughout the manuscript. A thorough review is necessary to enhance clarity and professionalism.
- The authors only consider a single dataset. Demonstrating the applicability of their approach across multiple publicly available datasets would strengthen the paper’s generalizability.
- The authors use the terms "movie" and "video" interchangeably, creating ambiguity regarding whether "movie" refers to video with audio or just video alone. This should be clearly defined.
- Main results, such as those in Table 1, should not be conflated with ablation studies to avoid confusion.
- The manuscript is missing comparisons to other identification/classification approaches, both within and across subjects, which would provide valuable context and relevance to the proposed method.

**Questions:**

1) I suggest creating figures with plots to visually represent the data. Each figure should have clear takeaways that are explicitly stated in their captions to guide the reader's understanding.

2) Why is the inter-movie accuracy typically higher than the intra-movie accuracy in Tables 1, 2, and 3? Given that inter-movie represents the held-out movie case, which is expected to be more challenging, this finding is counterintuitive and warrants further explanation.

3) It would be beneficial to apply the method to another dataset, ideally focusing on a single modality. This would help demonstrate the robustness and generalizability of the approach.

4) If feasible, I recommend breaking down the impact of the Vision Transformer (ViT) compared to the CLIP approach. This could involve substituting CLIP with a more conventional training and classification method (e.g., referencing works from the Kamitani and Gallant group over the past decade). Such a comparison would strengthen the case for the specific contributions related to subject canonical space mapping, rather than merely showcasing advanced training and classification techniques.

---

> ### Author Response · Authors · 2024-11-22
> **Rebuttal [1/3]**
>
> We thank the reviewer for highlighting concerns about readability. On reflection, we agree that we were overzealous in wanting to show everything. We have worked hard to pair back the paper, refine the use of language, and clarify our contributions.
>
> ### Weaknesses
>
> We organised this section into five main parts. The original review has 9 points of weakness. We highlight our response to individual points in bold.
>
> #### **1. Originality of the work**
>
> Our understanding is that the reviewer is asking for the paper to be contextualised relative to the MAE and videoMAE; however, we see these more as (paradigm-shifting) software engineering and image modelling innovations, whereas our key contributions lie in addressing the neuroscience constraints of the fMRI decoding problem.
>
> Ultimately, the cortex is a highly curved manifold – with distances between functional areas better calculated as geodesics along that surface than Euclidean distances in 3D. The improvements in the precision of inter-subject comparisons afforded through cortical surface processing have been expanded on in many papers `[1,2,3]`. However, classical neuroimaging analyses assume it is possible to map all data to a space in which brain regions (or functional areas) perfectly overlap – when, in reality, this is not the case (mappings are highly ill-posed). Learning-based frameworks have offered a potential solution for some time, but it was not until the SiT came along that there was any good way of modelling the complex properties of cortical imaging features across the surface. `[4]` benchmarked many existing surface convolutional frameworks for cortical phenotyping tasks but found they traded off feature expressivity against transformation equivariance. `[5]` tried to use surface CNNs to decode fMRI but found that the approach did not generalise to new subjects – probably because of this lack of transformation invariance.
>
> This is the first paper to address all of the engineering challenges and put this all together in a single unified **neuroscience-informed fMRI decoding framework**.  It is the reason that the model **generalises** to unseen brains - on what is a highly challenging data set for decoding (see more points on this in response to Q3 below) - and it is the reason we have no doubts that it would generalise to other more complex stimuli and problems.
>
> We have now expanded on these points in the contributions, related works and discussion sections of the paper (highlighted in blue).
>
> #### **2. On using retrieval for evaluation**
>
> We understand the attraction of reconstruction, but there are several reasons why we felt this was not the most meaningful metric to use for validation on this dataset. Our primary objective with this paper is to show that we could generalise decoding - to brains/individuals not seen during training - and for this we needed a large enough dataset (in terms of number of subjects) to model enough inter-subject variability (in cortical organisation).  We use the HCP 7T fMRI dataset with **174 participants**, which provides this but at the cost of much less densely sampled stimuli relative to datasets previously used for reconstruction: recording fMRI responses to short (1 - 4.3mins) *but diverse* clips of different movies across 174 different subjects (no repeats). By contrast, the NSD used by (e.g. Ozcelik 2023; Scotti 2023; 2024; Thual 2023) densely samples the responses of **8 individuals** to 8859 static images, shown 3x each *over 40 scanning sessions*. This trade-off of the number of participants relative to the richness of the stimuli they are shown is unavoidable due to the costs of scanning – we were therefore not aware of any large open datasets that did both. Certainly not those also benefiting from HCP functional processing pipelines.
>
> Under the constraints that we are working with much fewer data per subject, top-k CLIP retrieval remains a robust and well-validated metric – that, importantly, still forms a core component of the evaluation of recent reconstruction papers by Scotti (2023;2024) and Benchetrit (2023). In Figure 7, we showed some exemplary reconstruction from subjects not seen during training as evidence that our model encodes core semantic information from each scene. Following the reviewer’s requests, we add more examples to Appendix C (Figure C.9 and Figure C.10) and expand on our above justifications in the discussion of the paper  (highlighted in blue).

---

> > ### Author Response · Authors · 2024-11-22
> > **Rebuttal [2/3]**
> >
> > #### **3. Presentation of the results**
> >
> > On reflection, we agree that the density of results makes the paper difficult to read; however, we believe that this issue stems more from the presentation than the experimental setup.  In response, we have taken on board many of the reviewer’s suggestions:  **adding plots instead of some of the tables** (Figure 5); **using consistent and clear terminology**; **fixing notation and grammar**; and introducing **a new visualisation** to simplify the description of the experimental setup (Figure 2). In Figure 1, we rewrote the caption and increased the text size where possible. Many of these corrections are now highlighted in blue.
> >
> > The description of the entire pipeline is necessarily complicated, as it involves multiple models and two training phases, which has presented challenges with notation. Regardless, the reviewer is completely correct that we made mistakes,e.g., multiple uses of $T$ (for time and tokens). We believe we have now fixed all these and removed unnecessary equations and subscripts. The use of **terminology** was not always clear and consistent throughout the manuscript, so we have worked on this: specifically, to clarify the distinction between “movie scenes” and “movie clips” and intra vs inter sampling (more details below):
> >
> > #### **4. Inter/Intra movie sampling**
> >
> > One of the key issues was that Table 1 was trying to convey too much. In this experiment, “inter-/intra- movie sampling” did not describe performance within the training data set and the test set. Instead, it referred to the setup of CLIP retrieval: sampling negative pairs either from the same movie scene (intra, more challenging) or from different movie scenes (inter, easier). We agree with the reviewer that this terminology was unnecessarily confusing and detracts from the main message of the paper. Therefore, we have **simplified Table 1** to focus solely on the most difficult task and the most important results.
> >
> > Furthermore, we have replaced the inter/intra terminology with **soft/hard negatives** - borrowing terms from the retrieval literature `[6]`.  Specifically, *Intra-sampling* was replaced by **hard negatives** to reflect the greater challenge of disentangling between negatives sampled from the same movies (i.e. choosing between clips that are semantically more similar); and *inter-sampling* was replaced by **soft negatives**.
> >
> > #### **5. Comparison against state-of-the-art decoding models**
> >
> > In this paper, we compare our method against a ridge regression decoding baseline. Linear regression is a commonly used decoding technique used by state-of-the-art decoding frameworks – namely Thual (2023) and Ozcelik (2023). Other papers do employ non-linear regression (Scotti 2023;2024), which wasn’t benchmarked here. What was done – which hasn’t been done by any state-of-the-art frameworks we are aware of  –  was the functional alignment of all cortical maps using MSMall  (Robinson 2018) -  the HCP’s MSMall cortical surface registration software. This was shown by Glasser et al. 2016 (Nature)  -  to be fundamental to the development of their multimodal cortical areal parcellation; where the difference in functional overlap (across individuals) prior to MSMall and after is stark. That being said (and as discussed in our introduction and related works), there remain residual variations that such diffeomorphically-constrained mapping techniques cannot normalise; it is these that the SiT is modelling - significantly improving decoding performance in the process – where these findings align with a wealth of previous studies showing the topography (size and placement) of cortical areas is predictive of behavioural traits and IQ (e.g. Bijsterbosch 2018). In short, there are many reasons to suggest that our ridge regression baseline would perform extremely competitively against volumetric benchmarks such as Ozcelik (2023), and Scotti (2023;2024). To directly compare against these models would require us to roll back all of the image processing done by the HCP and map data back into the volume. As Coalson et al. 2018 `[3]` explains, this can only make things worse.

---

> > > ### Author Response · Authors · 2024-11-22
> > > **Rebuttal [3/3]**
> > >
> > > ### Questions
> > >
> > > #### **Q1. Improving Figures and adding plots**
> > >
> > > We have supplemented the results of Tables 2 and 3 with plots which we agree tell a much better story (Figure 5 and Figure C.5). We have also created a new Figure 2, which explains the setup for each experiment (Table 1 and Figure 5). All captions have been simplified. Brain regions are now annotated in Figure 3 to improve interpretability. Those changes are highlighted in blue.
> > >
> > > #### **Q2. Inter- VS Intra-movie accuracy**
> > >
> > > As stated above, it is actually the intra (now, **hard-negative**) sampling that is more difficult – because negative CLIP embeddings reflect movie clips with very similar semantic content. We have worked hard to clarify this through changing terminology and explaining in the text Section 4.3.
> > >
> > > #### **Q3. Generalisability to another dataset on a single modality**
> > >
> > > We agree that it will be important to evaluate the performance of our model on other datasets and tasks, such as the Narratives dataset, which records brain responses to audiobooks. However, in many ways, the 7T MRI movie dataset can be seen as a more challenging dataset for decoding since stimuli are sampled very sparsely (short and diverse movie scenes), but data is acquired across a very large cohort of individuals of subjects. In contrast, commonly used datasets, such as the Natural Scenes dataset, for decoding and reconstruction papers (e.g., Ozcelik et al., 2023; Scotti et al., 2023, 2024; Thual et al., 2023) involve densely sampled data from only eight subjects, presenting them with 8,859 static images repeated three times over 40 scanning sessions. Therefore, we have strong reasons to believe that if our model performs well on the diverse and large-scale HCP 7T movie-watching dataset, it is likely to generalise effectively to other problems.
> > >
> > > Note (also in response to R1) that our motivation for choosing the HCP 7T movie-watching dataset was driven by our objective to demonstrate the generalisation across subjects. For this, we needed enough examples from different brains to model individual cortical organisational variability in a generalisable way. Previous work, such as Glasser et al. 2016 (Nature) `[1]`, has shown that the HCP dataset indeed captures such individual variability, further supporting our choice. This trade-off - of the number of participants relative to the richness of stimuli - is unavoidable due to the costs of scanning, and we are not aware of any large open datasets that do both.
> > >
> > > #### **Q4. Impact of Vision Transformer (ViT) compared to the CLIP approach**
> > >
> > > We agree with the reviewer that this would be an interesting experiment to perform, but we prioritised improving the paper presentation and ran out of time for this. It is very difficult to apply semantic labelling to movie scenes from the HCP 7T fMRI movie watching (we have tried), but what we could do is set up an experiment which classifies which movie a scene comes from? This could be compared for the CLIP embeddings, the direct output of the vsMAE encoder, and the ridge regression (with and without CLIP). We would like to do this and add it to the appendix prior to any publication.
> > >
> > >
> > >
> > > ---
> > >
> > > [1] Glasser, M. F. et al. A Multi-modal Parcellation of Human Cerebral Cortex. Nature in press, (2016).
> > >
> > > [2] Glasser, M. F. et al. The Human Connectome Project’s neuroimaging approach. Nat. Neurosci. 19, (2016).
> > >
> > > [3] Coalson, T. S., Van Essen, D. C. & Glasser, M. F. The impact of traditional neuroimaging methods on the spatial localization of cortical areas. Proc. Natl. Acad. Sci. U. S. A. 115, E6356–E6365 (2018).
> > >
> > > [4] Fawaz, Abdulah, et al. "Benchmarking geometric deep learning for cortical segmentation and neurodevelopmental phenotype prediction." bioRxiv (2021): 2021-12.
> > >
> > > [5] Gu, Z., Jamison, K., Kuceyeski, A. & Sabuncu, M. Decoding natural image stimuli from fMRI data with a surface-based convolutional network. arXiv [cs.CV] (2022).
> > >
> > > [6] Unsupervised Learning of Visual Features by Contrasting Cluster Assignments, M.Caron et al 2020, NeurIPS 2020
> > >
> > > ---
> > > ---

---

> > > > ### Comment · Reviewer_vWQL · 2024-11-27
> > > >
> > > > I appreciate the authors' response and the notable revisions and clarifications made.
> > > > The paper appears much clearer to me now.
> > > >
> > > > A few remaining suggestions:
> > > > - While basic uncertainty estimates by std are indicated on the plots/tables, I couldn't find proper statistical significance assessments. Related, if the gains are argued for the average scores, perhaps SEM or CI be better stats to report.
> > > > - I believe that the main approach figure (Figure 1) can be further improved (barely noticed any changes) -- specifically any text labels or icons should approximately match the main text size, if not bigger; Acronyms would be easier to read if spelled out or appeared in a legend.

---

> > > > > ### Author Response · Authors · 2024-11-27
> > > > >
> > > > > We would like to thank the reviewer for their follow-up suggestions. These were two important points to address. Below, we outline the specific changes made in response to the reviewer’s feedback.
> > > > >
> > > > > 1. We have uploaded a revised version of Figure 1 with enlarged text to improve readability within the constraints of the available space. Although we initially attempted to incorporate a legend directly within the figure, the high number of notations made it unintelligible. To maintain clarity, we have ensured that all acronyms are fully spelled out in the figure’s caption.
> > > > >
> > > > > 2. Regarding uncertainty estimates, we have updated all relevant tables and plots to display 95% confidence intervals (CI) instead of standard deviations and ran paired t-tests with Bonferroni correction to compare the SiT and Ridge models. We made sure to recover the original data to verify our measurements and the normality assumption. We realise now that std was a very conservative uncertainty estimate, given our high sample size of 992 and 400 patients (respectively for experiment 2 and 3). We would like to thank again the reviewer, as we believe this strengthens our statement.
> > > > >
> > > > > These changes have been incorporated into Table 1 (main results for Experiment 1), Figure 5 (soft-sampling results for Experiments 2 and 3), Tables C1, C2, and C3 in the appendix (extensive results for Experiments 1, 2, and 3, respectively), and Figure C.5 (hard-sampling results for Experiments 2 and 3).

---

> > > > > > ### Comment · Reviewer_vWQL · 2024-11-27
> > > > > >
> > > > > > Thanks! Scores adjusted.

---

### Official Review · Reviewer_sizD · 2024-11-09

**Soundness:** 4
**Presentation:** 4
**Contribution:** 3
**Rating:** 8
**Confidence:** 4

**Summary:**

The authors present a novel approach to modeling cortical functional dynamics by leveraging Vision Transformers (ViTs) tailored for surface data analysis. Their method integrates a tri-modal alignment using CLIP, aligning audio, video, and fMRI data to enhance the retrieval of visual and auditory stimuli from cortical activity patterns and vice versa. The model demonstrates high within-subject accuracy, effectively identifying audio and video samples from just 3-second fMRI recordings. Additionally, the authors investigate attention maps that reveal distinct patterns associated with the brain's semantic and visual systems. They explore model's generalizability on (i) unseen subjects; (ii) unseen stimuli; and (iii) unseen stimuli within unseen subjects.

**Strengths:**

The authors:
- design a novel framework composed of SiTs and tri-modal CLIP alignment for brain decoding.
- evaluate the model on unseen subjects and unseen stimuli
- leverage attention maps providing interpretability of the model's behavior, and make the connection between the information extracted from attention heads and specific functional networks
- show the recontruction of visual stimuli improves through the tri-modal approach

**Weaknesses:**

- Limited to decoding 3-second clips, may not capture longer-term temporal dynamics
- Only tested on movie watching data, unclear how well it would generalize to other tasks/stimuli
- Requires high computational resources (multiple A100 GPUs)
- Limited diversity in the movie stimuli used for training

**Questions:**

What constraints or shortcomings resulted from training the models using only 3-second temporal windows? How might this impact the model's ability to capture longer-term temporal dynamics or more complex cognitive processes?

The caption for Figure 4 lacks clarity in identifying specific brain regions. It would be more informative to directly label or annotate the relevant areas (such as Broca's area, auditory cortex, sensorimotor cortex, visual areas, and the face area) on the attention map visualizations.

How does performance compare to other state-of-the-art brain decoding methods?

What are the privacy/ethical implications of being able to reconstruct visual experiences from brain data?

The study uses a fixed temporal lag of 6 seconds to account for the hemodynamic response. However, the hemodynamic response lag is known to vary across different brain regions. How might this fixed lag approach impact the model's ability to accurately capture temporal dynamics in regions with differing hemodynamic response functions?

---

> ### Author Response · Authors · 2024-11-22
> **Rebuttal [1/3]**
>
> We want to thank the reviewer for raising insightful points and questions that will help us further improve the manuscript; addressing specific points:
>
> ### Weaknesses
>
> #### 1. Limitations of using 3-second clips
>
> We agree with the reviewer that our choice to encode 3-second time windows would have its limitations -  especially if we later wish to model more complex cognitive processes. However, in practice, there is nothing stopping us from working with longer sequences:  time corresponds to the channel depth of each input patch - thus it has no impact on the complexity of the model.  And if need be, the embedding dimension $D$ could be increased to take into account more input frames.  In Appendix C, we added two figures, Figure C11 and Figure C12, reporting frame reconstruction results on longer time frames (respectively, **16 and 64 frames**), showing that this can be performed if need be and that the model still captures functional variability for individuals in longer time-windows.
> Here, we decided to work with 3-second clips for three reasons: (1) the average movie clip duration detected by PySceneDetect (defined as “shot changes in videos”) was about 3 seconds on average across most movies in fMRI sessions (see Table 2 in Appendix A.2); (2) the 3-second clip reconstruction with vsMAE showed the better qualitative and quantitative reconstruction results (Appendix C); (3) With longer time windows the number of non-overlapping movie-clips decreases a lot which limit further the pool of training samples which is already relatively limited for the HCP 7T dataset. The CLIP retrieval results and reconstruction using 3-second clip also has proved to be competitive in this setting.
>
> #### 2. Generalisation to tasks beyond movie-watching
>
> We agree that it will be important to evaluate the performance of our model on other datasets and tasks, such as the Narratives dataset `[2]`, which records brain responses to audiobooks. However, in many ways, the HCP 7T MRI movie dataset can be seen as a most challenging dataset for decoding since stimuli are sampled very sparsely (short and diverse movie scenes) but, data is acquired across a very large cohort of individuals of subjects. In contrast, commonly used datasets, such as the Natural Scenes dataset, for decoding and reconstruction papers (e.g., Ozcelik et al., 2023; Scotti et al., 2023, 2024; Thual et al., 2023) involve densely sampled data from only eight subjects, presenting them with 8,859 static images repeated three times over 40 scanning sessions. Therefore, we have strong reasons to believe that if our model performs well on the diverse and large-scale HCP 7T movie-watching dataset, it is likely to generalise effectively to other problems.
>
> Note (also in response to R1) that our motivation for choosing the HCP 7T movie-watching dataset was driven by our objective to demonstrate the generalisation across subjects. For this, we needed enough examples from different brains to model individual cortical organisational variability in a generalisable way.  Previous work, such as Glasser et al. 2016 (Nature), has shown that the HCP dataset indeed captures such individual variability, further supporting our choice.
>
>
> #### 3. Computational cost (A100 GPUs -> V100 GPUs)
>
> We apologise to the reviewers for making a mistake in the submission regarding the GPU model regarding the model of GPU used. For this study, we actually used ***V100s* GPUs** (which have 32Gb of memory, as previously indicated in Appendix B.4) and not *A100s* (80GB of memory). Nonetheless, we acknowledge the reviewer’s concern that computational requirements remain high.  The use of multiple GPUs was necessary for distributed training, which enhances performance, particularly for the tri-modal CLIP self-supervision training, but not essential for running our pipeline. Of note, we will provide our custom code for CLIP parallelisation, as we could not find any easy-to-use resources for this. In some ways, we think that high computational demands are inevitable since human brain functional dynamics are complex. However, our approach offers key advantages through working on the cortical surface instead of the volume – since it downsamples the data in a biologically-informed way that preserves the resolution and spatial organisation of the fMRI activation patterns. This allows our pipeline to run on a single 24 GB RTX3090 GPU (or equivalent) with a batch size 32, thereby enhancing scalability and accessibility.

---

> > ### Author Response · Authors · 2024-11-22
> > **Rebuttal [2/3]**
> >
> > #### 4. Limited diversity of the HCP 7T dataset
> >
> > We understand the point of the reviewer regarding the diversity of the HCP 7T dataset. However, this dataset comprises a large variety of movie genres, with a mix of blockbusters and independent movies. For blockbusters, the movie scenes were carefully selected based on various criteria (scene dynamics, motion changes, luminance and colour contrast), making them highly diverse while relatively short `[1]`. For instance, some scenes feature long dialogue with minimal motion, while others involve intense action with little dialogue. For these reasons, we believe that the capabilities of our model to retrieve and reconstruct such diverse stimuli just should indicate strongly that the model would generalise to other types of (more densely sampled) complex stimuli.
> >
> > ### Questions
> >
> > #### 1. Comparison against state-of-the-art decoding models
> >
> > In this paper, we compare our method against a ridge regression decoding baseline. Linear regression is a commonly used decoding technique used by state-of-the-art decoding frameworks – namely Thual (2023) and Ozcelik (2023). Other papers do employ non-linear regression (Scotti 2023;2024), which wasn’t benchmarked here. What was done – which hasn’t been done by any state-of-the-art frameworks we are aware of  –  was the functional alignment of all cortical maps using MSMall  (Robinson 2018) -  the HCP’s MSMall cortical surface registration software. This was shown by Glasser et al. 2016 (Nature)  -  to be fundamental to the development of their multimodal cortical areal parcellation; where the difference in functional overlap (across individuals) prior to MSMall and after is stark. That being said (and as discussed in our introduction and related works), there remain residual variations that such diffeomorphically-constrained mapping techniques cannot normalise; it is these that the SiT is modelling - significantly improving decoding performance in the process – where these findings align with a wealth of previous studies showing the topography (size and placement) of cortical areas is predictive of behavioural traits and IQ (e.g. Bijsterbosch 2018). In short, there are many reasons to suggest that our ridge regression baseline would perform extremely competitively against volumetric benchmarks such as Ozcelik (2023), and Scotti (2023;2024). To directly compare against these models would require us to roll back all of the image processing done by the HCP and map data back into the volume. As Coalson et al. 2018  `[3]` explains, this can only make things worse.
> >
> > #### 2. Clarity in attention maps
> >
> > Thank you for your excellent suggestion. We have now annotated the maps with commonly recognised brain areas, including early and association auditory cortices, Broca’s area, somatosensory and motor cortices, visual cortices, and the face detection area (Figure 4).
> >
> > #### 3. Privacy and ethics
> >
> > This is an excellent point. The potential for decoding thoughts from individual brains, without the need for training data, should raise ethical concerns of misuse or misinterpretation (one potential example we would worry about would be use in criminal trials). We would agree that it is important to start this conversation in this field now.
> >
> > That being said, such technology offers significant power for good: Neurofeedback and Brain-Computer Interface (BCI) paradigms are emerging as promising fields of research that could revolutionise patient care and tailor treatments/surgical interventions to individual brains, with some already working applications that, e.g., stimulate leg movements in tetraplegic patients from cortical recordings `[4]`.
> > It is important to stress that we are a very long way from being able to precisely decode the complex thoughts of an individual human brain, while still dependent on MRI imaging, which is not portable and not accessible by anyone. Considering the importance of this discussion we have added some of this detail to the discussion.
> >
> > #### 4. Temporal Lag
> >
> > Thank you very much to the reviewer for raising this very interesting point. Indeed, in this study we used a fixed temporal lag, as it showed to be the most predictive of cortical activation, by regressing fMRI time series from the video latent embeddings (Appendix C.8). However, for future studies, it might be possible to introduce ‘temporal embeddings’; a set of learnable parameters per patch to weigh the contribution of each frame, in order to account for the spatial variability of the temporal dynamics.

---

> > > ### Author Response · Authors · 2024-11-22
> > > **Rebuttal [3/3]**
> > >
> > > ---
> > >
> > > 1. Cutting, J. E., Brunick, K. L., & Candan, A. (2012). Perceiving event dynamics and parsing Hollywood films. Journal of Experimental Psychology: Human Perception and Performance, 38(6), 1476–1490. https://doi.org/10.1037/a0027737
> > > 2. Nastase, S. A. et al. The ‘Narratives’ fMRI dataset for evaluating models of naturalistic language comprehension. Sci. Data 8, 250 (2021).
> > > 3. Coalson, Timothy S., David C. Van Essen, and Matthew F. Glasser. "The impact of traditional neuroimaging methods on the spatial localization of cortical areas." Proceedings of the National Academy of Sciences 115.27 (2018): E6356-E6365.
> > > 4. Walking naturally after spinal cord injury using a brain–spine interface, H.Lorach et al 2023, Nature
> > > ---
> > > ---

---

> > > > ### Author Response · Authors · 2024-12-02
> > > > **Looking forward to your feedback**
> > > >
> > > > Dear Reviewer sizD, the author/reviewer discussion phase is about to end (December 2nd). Do you have any additional feedback given our response? Thank you!

---

> > > > > ### Comment · Reviewer_sizD · 2024-12-02
> > > > > **Post-rebuttal**
> > > > >
> > > > > The authors have satisfactorily addressed most of my questions through their responses and revisions. The remaining points are minor and do not impact the paper's overall contribution. I maintain my original score as appropriate.

---

### Official Review · Reviewer_zoi6 · 2024-11-11

**Soundness:** 3
**Presentation:** 3
**Contribution:** 3
**Rating:** 6
**Confidence:** 3

**Summary:**

This paper proposes a new technique for decoding brain signals that is generalizable across participants. They integrate surface vision transformers (SiTs) with multi-modal contrastive learning to decode audio and visual stimuli from cortical fMRI data. They demonstrate the efficacy of their approach on the HCP movie watching database and show superior decoding performance. The model shows strong performance not only on the same subjects and stimuli seen during training but also on new subjects and previously unseen stimuli

**Strengths:**

- The idea of multimodal alignment of vision, audition and fMRI through contrastive learning seems novel and promising
- The study demonstrates that the model can generalize across unseen subjects and stimuli, which is a crucial aspect of developing robust brain decoding methods.
- The paper builds on established techniques in brain decoding by introducing more advanced architectures, like surface vision transformers (SiTs) which are better able to capture long-range structure in surface imaging data

**Weaknesses:**

- Comparison against alternative architectures like 3D CNNs etc is lacking in this work, which makes it harder to understand the contribution of the Surface vision transformer
-  Quantitative evaluations of the reconstructed visual clips against other state of the art decoding models is also currently lacking in those work. At the very least, the authors should include more examples of reconstructed clips
- Model performance is presented using retrieval tests which are much easier than reconstructions; Given that most studies nowadays can perform decent decoding with the help of powerful generative models, this should atleast be discussed.

**Questions:**

- During inference, why not use the full test set for retrieval? The size of the set clearly has an impact on the retrieval performance. Why do the authors choose a lower set size for the auditory domain?
- Can the authors comment on the biological plausibility of the encoder architectures used for stimulus encoding - eg wav2vec which works directly with raw audio waveforms instead of cochleograms/spectrograms may not be the best choice for reformatting auditory signals into brain-aligned representations

---

> ### Author Response · Authors · 2024-11-22
> **Rebuttal [1/2]**
>
> We want to thank the reviewer for their insightful remarks and questions; addressing specific points:
>
> ### Weaknesses
>
> #### 1. Comparison against 3D CNNs
>
> Unfortunately, it is not possible to make a direct comparison against 3D CNNs, since working with 4D 7T fMRI data at full resolution ($1.6mm^3$; 130\*130 slice resolution) is computationally infeasible. For this reason, state-of-the-art methods for 3D modelling of fMRI data, such as Kim et al. 2023 `[1]` - which uses Swin Transformers - substantially downsample the 3D volumes through patching: image resolution 96\*96\*96;patch resolution 16\*16\*16. This has the effect of merging signal (from grey matter voxels) with noise (from white matter and CFS voxels). Moreover, modelling signals in 3D is not recommended since Euclidean distances misrepresent the true geodesic distances between functional activations along the cortical sheet. For example, one might assume that a convolutional kernel applied to low-resolution volumetric fMRI would average signal across opposing banks of a sulcus – however, there is ample evidence to show that different functional areas are found at these locations; thus, the convolutional operation would confuse (merge) signal from different functional processes; making it less suitable for fine-scale decoding of stimuli `[2, 3]`.
>
> To our knowledge, all papers that report fMRI decoding from volumetric data treat voxels as independent signals and process fMRI using linear or non-linear regression modules; typically, only using voxels from a masked region of the posterior cortex (Ozcelik et al. 2023; Scotti et al. 2023; 2024). This is one of the key advantages of working on the cortical surface – since it downsamples the data in a biologically informed way that preserves the resolution of the cortical signal and spatial-organisation of the fMRI activation patterns. The HCP go even further to functionally align cortical imaging data using MSMall `[4,5]`, showing that this considerably improves the overlap of functional activations across brains `[2,3]`.
>
> ### 2-3. Reconstructions results & using retrieval for evaluation
>
> We understand the attraction of reconstruction, but there are several reasons why we felt this was not the most meaningful metric to use for validation on this dataset. Our primary objective with this paper is to show that we could generalise decoding - to brains/individuals not seen during training - and for this we needed a large enough dataset (in terms of number of subjects) to model enough inter-subject variability (in cortical organisation).  We use the HCP 7T fMRI dataset with **174 participants**, which provides this but at the cost of much less densely sampled stimuli relative to datasets previously used for reconstruction: recording fMRI responses to short (1 - 4.3mins) *but diverse* clips of different movies across 174 different subjects (no repeats). By contrast, the NSD used by (e.g. Ozcelik 2023; Scotti 2023; 2024; Thual 2023) densely samples the responses of **8 individuals** to 8859 static images, shown 3x each *over 40 scanning sessions*. This trade-off of number of participants relative to the richness of the stimuli they are shown is unavoidable due to the costs of scanning – we were therefore not aware of any large open datasets that did both. Certainly not those also benefitting from HCP functional processing pipelines.
>
> Under the constraints that we are working with much fewer data per subject, top-k CLIP retrieval remains a robust and well-validated metric – that, importantly, still forms a core component of the evaluation of recent reconstruction papers by Scotti (2023;2024) and Benchetrit (2023). In Figure 7, we showed some exemplary reconstruction from subjects not seen during training as evidence that our model encodes core semantic information from each scene. Following the reviewer’s requests, we add more examples to Appendix C (Figure C.9 and Figure C.10) and expand on our above justifications in the discussion of the paper (highlighted in blue in the new submission).

---

> > ### Author Response · Authors · 2024-11-22
> > **Rebuttal [2/2]**
> >
> > ### Questions
> >
> > #### 1. Inference - test size
> >
> > At inference time, we opted to use 64 negatives to accommodate the variability in the duration of each fMRI session. Since each fMRI session comprises different movies, the total duration and, consequently, the number of non-overlapping 3-second movie clips varies between sessions. Since each experiment compares across fMRI sessions and subjects, we decided to use a fixed number of negatives (corresponding approximately to 1/5 of the total number of video clips from each of the four fMRI sessions). Moreover, to account for the variability in model performance across movie scenes (some considered more challenging than others), the inference was repeated 100 times for each test subject) using different reference clips and corresponding sets of negatives. The final top-k accuracy for each experiment is calculated by averaging across repeats (for all test subjects).
> > For audio retrieval, we reduced the test set size because the number of ‘meaningful’ audio samples is significantly lower than that of video samples; many audio samples are either completely noisy or contain no sound.
> >
> > #### 2. Multimodal architectures used for stimulus encoding
> >
> > We would like to thank the reviewer for bringing this issue to our attention. We decided to use the wav2vec 2.0 model to extract audio embeddings from the movie clips, as this model had been used in previous significant studies focusing on speech encoding/decoding `[6]`. However, we acknowledge the reviewer's remark that encoding spectrograms or cochleograms may be better aligned with auditory brain-signal representation, and we will certainly take this into account in our future works. We note that despite using raw waveform, results in Table 1 still demonstrate the benefit of using all three modalities: adding $\mathcal{A}$ marginally improves the performance for video decoding $f_{MRI} \rightarrow \mathcal{V}$ and using $\mathcal{A}$ largely improves the decoding $f_{MRI} \rightarrow \mathcal{V}$ of $SiT$ compared to the Ridge model.  This seems to indicate that, even with raw audio waveforms, the multimodal approach effectively leverages complementary information to improve retrieval accuracy.
> >
> >
> > ---
> >
> > 1. SwiFT: Swin 4D fMRI Transformer, PY Kim et al 2023, NeurIPS 2023
> > 2.  The minimal preprocessing pipelines for the Human Connectome Project, M.Glasser et al 2013, NeuroImage
> > 3. The impact of traditional neuroimaging methods on the spatial localization of cortical areas, T.Coalson et al 2018, PNAS
> > 4. Robinson, E. C. et al. MSM: A new flexible framework for multimodal surface matching. Neuroimage 100, 414–426 (2014).
> > 5. Robinson, E. C. et al. Multimodal surface matching with higher-order smoothness constraints. Neuroimage 167, (2018).
> > 6. Toward a realistic model of speech processing in the brain with self-supervised learning, J.Mille et al 2023, NeurIPS 2023
> >
> >
> > ---
> > ---

---

> > > ### Author Response · Authors · 2024-12-02
> > > **Looking forward to your feedback**
> > >
> > > Dear Reviewer zoi6, the author/reviewer discussion phase is about to end (December 2nd). Do you have any additional feedback given our response? Thank you!

---

### Meta-Review · Area_Chair_7akB · 2024-12-12

**Metareview:**

This submission contributes a brain decoding method that generalizes across participants, using fMRI data projected on the cortical surface and modeled with surface vision transformers and contrastive learning. The submission generated interest from the reviewers with the learning across subject seen as important and the method interesting. One limitation brought forward was the validation of the method only in the context of movie-watching data.

The abstract positions the contributions in the context of brain compute interface, but the work is performed on fMRI. This positioning is not a good one: fMRI requires an MRI scanner which cannot be used for day-to-day decoding, and thus not for BCIs.

**Additional Comments On Reviewer Discussion:**

There was a thorough discussion with back and forth between authors and reviewers, though one reviewer did not follow up on the authors answers.

The discussion led to improving the clarity of the manuscript as well as better understanding of the contributions.

---

### Decision · Program_Chairs · 2025-01-22

Accept (Poster)